# Seasonal characteristics, formation mechanisms and source origins of PM2.5 in two megacities in Sichuan Basin, China

Huanbo Wang[1,2], Mi Tian[1], Yang Chen[1], Guangming Shi[1], Yuan Liu[1], Fumo Yang[1,2,3,4*], Leiming Zhang[5], Liqun Deng[6], Jiayan Yu[7], Chao Peng[1], and Xuyao Cao[1]

[1]Research Center for Atmospheric Environment, Chongqing Institute of Green and Intelligent Technology, Chinese Academy of Sciences, Chongqing, 400714, China

[2]School of Urban Construction and Environmental Engineering, Chongqing University, Chongqing, 400044, China

[3]Center for Excellence in Regional Atmospheric Environment, Institute of Urban Environment, Chinese Academy of Sciences, Xiamen, 361021, China

[4]Yangtze Normal University, Chongqing, 408100, China

[5]Environment and Climate Change Canada, Toronto, Canada

[6]Sichuan Academy of Environmental Sciences, Chengdu, 610041, China

[7]Chongqing Environmental Monitoring Center, Chongqing 401147, China

*Correspondence to:* Fumo Yang (fmyang@cigit.ac.cn)

**Abstract.** To investigate the characteristics of $PM_{2.5}$ and its major chemical components, formation
mechanisms, and geographical origins in the two megacities, Chengdu (CD) and Chongqing (CQ), in
Sichuan Basin of southwest China, daily $PM_{2.5}$ samples were collected simultaneously at one urban site
in each city for four consecutive seasons from autumn 2014 to summer 2015. Annual mean
concentrations of $PM_{2.5}$ were $67.0 \pm 43.4$ and $70.9 \pm 41.4$ $\mu g\ m^{-3}$ at CD and CQ, respectively. Secondary
inorganic aerosols (SNA) and organic matter (OM) accounted for 41.1% and 26.1% of $PM_{2.5}$ mass at
CD, and 37.4% and 29.6% at CQ, respectively. Seasonal variations of $PM_{2.5}$ and major chemical
components were significant, usually with the highest mass concentration in winter and the lowest in
summer. Daily $PM_{2.5}$ concentration exceeded the national air quality standard on 30% of the sampling
days at both sites, and most of the pollution events were at the regional scale within the basin formed
under stagnant meteorological conditions. The concentrations of carbonaceous components were higher
at CQ than CD, likely partially caused by emissions from the large amount of motorcycles and spraying
process during automobile production in CQ. Heterogeneous reactions probably played an important
role in the formation of $SO_4^{2-}$, while both homogeneous and heterogeneous reactions contributed to the
formation of $NO_3^-$. Geographical origins of emissions sources contributing to high $PM_{2.5}$ masses at both
sites were identified to be mainly distributed within the basin based on potential source contribution
function (PSCF) analysis.

## 1 Introduction

Fine particles (PM2.5, particulate matter with an aerodynamic diameter smaller than 2.5 μm) have adverse effects on human health (Anderson et al., 2012;Lepeule et al., 2012;Taus et al., 2008), deteriorate air quality (Zhang et al., 2008;Paraskevopoulou et al., 2015), reduce atmospheric visibility (Fu et al., 2016;Cao et al., 2012;Baumer et al., 2008), impact climate (Ramanathan and Feng, 2009;Hitzenberger et al., 1999;Mahowald, 2011), and affect ecosystem (Larssen et al., 2006). In the past two decades, China has experienced serious PM2.5 pollution due to the rapidily incresing energy consumption through econmic development, industrialization and urbanization (Tie and Cao, 2009). The National Ambient Air Quality Standards (NAAQS) for PM2.5 was promulgated by the Chinese government in 2012, and strict strategies have been implemented nationwide, e.g. controling $SO_2$ emissions by installing desulphurization system in coal-fired power plants and conversion of fuel to natural gas (Lu et al., 2011), mitigating $NO_x$ emissions through traffic restrictions, and reducing biomass burning through straw shredding. Despite these efforts, there are still many cities that have not yet met the current NAAQS (Tao et al., 2017). According to the '2013-2015 Reports on the State of Environment of China', annual mean concentration of PM2.5 in 74 major cities across China was 72, 64, and 50 μg m$^{-3}$ in 2013, 2014 and 2015, respectively, and only 4.1%, 12.2% and 22.5% of the monitored cities met the NAAQS (35 μg m$^{-3}$).

Previous studies showed that Beijing-Tianjin-Hebei area (BTH), Yangtze River Delta (YRD), Pearl River Delta (PRD), and Sichuan Basin were the four main regions in China with severe aerosol pollution (Tao et al., 2017). While many studies have been conducted in BTH, PRD and YRD regions to understand the general characteristics of PM2.5 and its chemical components, formation mechanism, and sources (Ji et al., 2016;Li et al., 2015;Quan et al., 2015;Tan et al., 2016;Yang et al., 2015;Zhang et al., 2013;Zhao et al., 2015;Zhao et al., 2013a;Cheng et al., 2015;Zheng et al., 2015a;Yang et al., 2011a), only a few studies have focused on the Sichuan Basin (Tao et al., 2014;Tian et al., 2013;Yang et al., 2011b). Covering an area of 260,000 km$^2$ and with a population of around 100 million, the Sichuan Basin is the most populated basin in China. It is a subtropical expanse of low hills and plains and is completely encircled by high mountains and plateaus. It is also characterized by persistently high relative humidity and extremely low wind speeds all the year-round (Guo et al., 2016;Chen and Xie, 2013). The characteristics of PM2.5 in the Sichuan Basin are supposed to be very different from those in eastern coastal China (i.e. PRD and YRD) and North China Plain (i.e. BTH) due to the special

topography and meteorological conditions, besides emission sources, in the basin. Furthermore, the terrain in the two megacities is also distinct from each other significantly, i.e., Chongqing is a mountainous city lying on the eastern margin of the basin while Chengdu is a flat city on the western margin of the basin. Therefore, there is a great interest in comparing the chemical components of $PM_{2.5}$ and characterizing pollution episodes between the two cities.

The present study aims to fill this gap by measuring chemically-resolved $PM_{2.5}$ in Chengdu and Chongqing in four consecutive seasons during 2014-2015. The main objectives are to: (1) characterize $PM_{2.5}$ mass and major chemical components in urban environemnts of Chengdu and Chongqing; (2) compare $PM_{2.5}$ chemical compositions under different pollution levels and identify major chemical components responsible for long-lasting $PM_{2.5}$ pollution episodes in winter; (3) explore the possible formation mechanism of the secondary aerosols; and (4) reveal the geographical source regions contibuting to the high $PM_{2.5}$ levels through PSCF analysis. Knowledge gained in this study provides scientific basis for making future emission control plocies aiming to allivating heavy $PM_{2.5}$ pollution in this unique basin.

## 2 Methodology

### 2.1 Sampling sites

$PM_{2.5}$ samples were collected at two urban sites, one in Chengdu and another in Chongqing, the two largest cities in Sichuan Basin, southwest China. The two sampling sites are located 260 km apart (Fig. 1). The sampling site in Chengdu (CD) is located on the roof of a sixth floor building in the Sichuan Academy of Environmental Science (104º4′ E, 30º37′ N) with no large surrounding industries but heavy traffic. The closest main road (Renmin South road of Chengdu) is about 20 m east of the samling site. The sampling site in Chongqing (CQ) is located on the rooftop of Chongqing Monitoring Center (106º30′ E, 29º37′ N). The highway G50 is 250 m away from this sampling site. The two selected sampling sites are considered to represent typical urban environments in their respective cities (Tao et al., 2014;Chen et al., 2017).

### 2.2 Sample collection

Daily (23-h) integrated $PM_{2.5}$ samples were collected in four months, each in a different season: autumn (23 October to 18 November, 2014), winter (6 January to 2 February, 2015), spring (2 to 29 April, 2015), and summer (2 to 30 July, 2015). At both sites, $PM_{2.5}$ samples were collected in parallel on Teflon filters (Whatman Corp., 47 mm) and quartz filters (Whatman Corp., 47 mm). At CD site, $PM_{2.5}$

sampling was carried out using a versatile air pollutant sampler (Wang et al., 2017). One channel was
used to load PM$_{2.5}$ sample on Teflon filter for mass and trace elements anlysis and the other one was
equipped with quatz filter for water-soluble inorganic ions and carbonaceous components analysis. The
sampler was running at 15 L min$^{-1}$ for each channel. At CQ site, a low-volume aerosol sampler (BGI
Corp., frmOmni, USA) operating at a flow rate of 5 L min$^{-1}$ was used to collect PM$_{2.5}$ samples on
Teflon filter, and another sampler (Thermo Scientific Corp. Partisol 2000i, USA) with a flow rate of
16.7 L min$^{-1}$ was used to collect PM$_{2.5}$ samples on quartz filter. A total of 112 samples and 8 field
blanks, nearly equally distributed in the four seasons, were collected at each site during the campaign.
In addition, three lab blank filters in each campaign were stored in a clean Petri slides in the dark and
analysed in the same ways as the collected samples to evaluate the background contamination.
Before sampling, all the quartz filters were preheated at 450°C for 4 h to remove the organic
compounds. All sampled filters were stored in clean Petri slides in the dark and at -18°C until analysis
to prevent the evaporation of volatile compounds. Before and after sample collection, all the Teflon
filters were weighted at least three times using an microbalance (Sartorius, ME 5-F, Germany) after
their stabilization for 48 h under controlled conditions (temperature: 20~23°C, relative humidity:
45~50%). Differences among replicate weights were mostly less than 15 µg for each sample.
**2.3 Chemical analysis**
For the analysis of water-soluble inorganic ions, a quarter of each quartz filter was first extracted using
ultrapure water in an ultrasonic bath for 30 min, and then filtered through a 0.45 μm pore syringe filter.
Anions ($SO_4^{2-}$, $NO_3^-$ and $Cl^-$) and cations ($Na^+$, $NH_4^+$, $K^+$, $Mg^{2+}$ and $Ca^{2+}$) were determined using ion
chromatograph (Dionex Corp., Dionex 600, USA). Anions were separated using AS11-HC column with
30 mM KOH as an eluent at a flow rate of 1.0 ml min$^{-1}$. Cations were determined using CS12A column
with 20 mM MSA (methanesulfonic acid) at a flow rate of 1.0 ml min$^{-1}$. Individual standard solutions of
all investigated anions and cations (1000 mg L$^{-1}$, o2si, USA) were diluted to construct the calibration
curves. The correlation coefficients of the linear regression of the standard curves were all above 0.999.
Field blanks were prepared and analyzed together with the samples and then subtracted from the
samples. The concentrations of the water-soluble inorganic ions in the field blanks were in the range of
0.008-0.13 μg m$^{-3}$. The relative standard deviation of each ion was better than 8% for the reproducibility
test.
Organic carbon (OC) and elemental carbon (EC) were measured by thermal-optical reflectance

(TOR) method using a DRI OC/EC analyzer (Atmoslytic Inc., USA). The methodology for OC/EC analysis was based on TOR method as described in Chow et al. (2007). For calibration and quality control, measurement with filter blank, standard sucrose solution and replicate analysis were performed. Blank corrections were performed by subtracting the blank values from the sampled ones. The concentration of EC in field blanks was zero while OC was below 0.7 $\mu g$ C $cm^{-2}$. The repeatability was better than 15%.

The elements including Al, Si, Ca, Fe, and Ti were analyzed on Teflon filter using X-ray fluorescence analyzer (Epsilon 5ED-XRF, PAN'alytical Corp., Netherlands), the QA/QC procedures of the XRF analysis have been described in Cao et al. (2012). The gaseous species were continuously measured by a set of online gas analyzers, including EC9850 $SO_2$ analyzer, 9841 $NO/NO_2/NO_x$ analyzer, 9830 CO analyzer, and 9810 $O_3$ analyzer (Ecotech Corp., Australia) at CD, and Thermo 42i $NO/NO_2/NO_x$ analyzer, 43i $SO_2$ analyzer, 48i CO analyzer, and 49i $O_3$ analyzer (Thermo Scientific Corp., USA) at CQ. The mass concentrations of $PM_{2.5}$ were automatically measured by online particulate monitor instruments (BAM1020, Met one Corp., USA, at CD and 5030 SHARP, Thermo Scientific Corp, USA, at CQ). Hourly meteorological parameters, including ambient temperature (T), relative humidity (RH), wind speed (WS) and direction, barometric pressure (P), and solar radiation (SR) were obtained from an automatic weather station (Lufft Corp. WS501, Germany) at each site. Hourly precipitation data were recorded at the nearest weather station operated by China Meteorology Administration (http://www.weather.com.cn/). Planetary boundary layer height (PBLH) was obtained from the HYSPLIT model (http://ready.arl.noaa.gov/HYSPLIT.php).

**2.4 Data analysis**

The EC-tracer method has been widely used to estimate SOC (Turpin and Lim, 2001;Castro et al., 1999), which can be expressed as

$$POC=(OC/EC)_{prim} \times EC \qquad (1)$$

$$SOC=OC-POC \qquad (2)$$

Where POC, SOC and OC represent the estimated primary OC, secondary OC and measured total OC, respectively. $(OC/EC)_{min}$ was simplified as the $(OC/EC)_{prim}$ to estimate SOC in this study. $(OC/EC)_{min}$ was 2.4, 2.6, 1.6 and 2.2 in autumn, winter, spring and summer at CD, respectively, and 1.9, 2.8, 1.1 and 1.5 at CQ. The estimated SOC was only an approximation with uncertainties, e.g., from influence of biomass burning (Ding et al., 2012).

The coefficient of divergence (COD) has been used to evaluate the spatial similarity of chemical compositions at different sites (Wongphatarakul et al., 1998;Qu et al., 2015), which is defined as

$$COD_{jk} = \sqrt{\frac{1}{p}\sum_{1}^{p}(\frac{x_{ij} - x_{ik}}{x_{ij} + x_{ik}})^2} \qquad (3)$$

Where $x_{ij}$ and $x_{ik}$ represent the average concentration for a chemical component $i$ at site $j$ and $k$, respectively, $p$ is the number of chemical components. Generally, a COD value lower than 0.2 indicates a relatively similarity of spatial distribution.

**2.5 Geographical origins of PM$_{2.5}$**

72-h air mass back trajectories were generated based on the Hybrid Single Particle Lagrangian Integrated Trajectory (HYSPLIT) model using 0.5°×0.5° meteorological data for the period of October 2014 to July 2015 when PM$_{2.5}$ measurements were made at both sites. Four trajectories at 04:00, 10:00, 16:00, and 22:00 UTC every day with the starting height of 300 m above ground level were calculated (Squizzato and Masiol, 2015).

PSCF is substantially a conditional probability that trajectories with pollutant concentrations larger than a given criterion passed through a grid cell ($i,j$) (Ashbaugh et al., 1985;Polissar et al., 1999), that means a grill cell ($i,j$) with high PSCF values are mostly potential source locations of pollutants. PSCF is defined as follows,

$$PSCF_{ij} = \frac{m_{ij}}{n_{ij}} \qquad (4)$$

Where $n_{ij}$ is the total number of endpoints falling in the grid cell ($i,j$) and $m_{ij}$ denotes the number of endpoints that are associated with samples exceeding the threshold criterion in the same cell. To reduce the PSCF uncertainties associated with small $n_{ij}$ values, weighting function was adopted as follows,

$$W_{ij} = \begin{cases} 1.0 & 3n_{ave} < n_{ij} \\ 0.7 & 1.5n_{ave} < n_{ij} \le 3n_{ave} \\ 0.42 & n_{ave} < n_{ij} \le 1.5n_{ave} \\ 0.2 & n_{ij} \le n_{ave} \end{cases} \qquad (5)$$

Where $n_{ave}$ is the average number of endpoints in each grid cell.

The trajectories coupled with daily pollutants concentrations were used for PSCF analysis, with the threshold criterion in PSCF analysis being set at the upper 50% of PM$_{2.5}$ and other pollutants. The trajectory covered area was in the range of 20-45° N and 90-120° E and divided into 0.5°×0.5° grid cells.

## 3 Results and discussion

### 3.1 PM$_{2.5}$ mass concentration and chemical composition

#### 3.1.1 Overview

Table 1 presents seasonal and annual mean concentrations of PM$_{2.5}$ and its major chemical components at CD and CQ during the sampling periods. Daily PM$_{2.5}$ ranged from 11.6 to 224.7 μg m$^{-3}$ with annual average being 67.0 ± 43.4 μg m$^{-3}$ at CD and 70.9 ± 41.4 μg m$^{-3}$ at CQ, which were about two times the NAAQS annual limit. Secondary inorganic aerosol (SNA, the sum of SO$_4^{2-}$, NO$_3^-$ and NH$_4^+$) and carbonaceous species together represented more than 70% of PM$_{2.5}$ mass at both sites (Fig. 2). The annual mean concentrations of SNA were 27.6 μg m$^{-3}$ at CD and 26.5 μg m$^{-3}$ at CQ, contributing 41.1% and 37.4% to PM$_{2.5}$ mass, respectively. SO$_4^{2-}$, NO$_3^-$ and NH$_4^+$ accounted for 16.8%, 13.6% and 10.8%, respectively, of PM$_{2.5}$ mass at CD, and 17.2%, 10.9% and 9.2%, respectively, at CQ. Organic matters (OM), estimated from OC using a conversion factor of 1.6 to account for other elements presented in organic compounds (Turpin and Lim, 2001), were the most abundant species in PM$_{2.5}$, accounting for 26.1% and 29.6% of PM$_{2.5}$ mass at CD and CQ, respectively. In contrast, EC only comprised of around 6% at both sites. The annual mean concentrations of OC and EC were 20% and 25%, respectively, and were higher at CQ than CD. The annual mean concentration of fine soil (FS), calculated by summing the oxides of major crustal elements, i.e., Al$_2$O$_3$, SiO$_2$, CaO, FeO, Fe$_2$O$_3$, and TiO$_2$ (Huang et al., 2014), was 6.7 μg m$^{-3}$ (9.5% of PM$_{2.5}$ mass) at CQ. It is noted that this was about two times that at CD (3.8 μg m$^{-3}$, 5.7% of PM$_{2.5}$ mass). The minor components such as K$^+$ and Cl$^-$ constituted less than 5% of PM$_{2.5}$. The unaccounted portions of PM$_{2.5}$ reached 18.3% at CD and 15.3% at CQ, which were likely related to the uncertainties in the multiplication factors used for estimating OM and FS, other unidentified species, and measurement uncertainties.

#### 3.1.2 Seasonal variations

Figure 3 shows the seasonal variations in mass concentrations of PM$_{2.5}$ and its major chemical components at CD and CQ. Seasonal variations of any pollutants were influenced by the seasonal variations in source emission intensities, atmospheric processes and meteorological conditions. Unlike in northern China, there were no extensive coal combustion or wood burning for domestic heating in winter due to the warm temperature (around 10°C on average) in the Sichuan Basin, hence atmospheric processes and meteorological conditions played vital roles in the seasonal variations of PM$_{2.5}$. On a seasonal basis, PM$_{2.5}$ mass was the highest in winter at both CD and CQ, which was 1.8-2.5 times of those in the other

seasons. In contrast, its seasonal differences among the other three seasons were generally small, i.e., less than 40%. Stagnant air conditions with frequent calm winds and low planetary boundary layer heights were the major causes of the highest $PM_{2.5}$ mass in winter (Table 1) (Liao et al., 2017;Chen and Xie, 2013;Li et al., 2017b).

All the major $PM_{2.5}$ components except FS followed the seasonal pattern of $PM_{2.5}$ mass with subtle differences. The highest FS concentrations were observed in spring at both sites. The relatively high wind speed and lower RH in spring were conducive for re-suspension of crustal dust and resulted in higher FS concentrations. In addition, frequent spring dust storms originated in the northwestern China was able to reach Sichuan Basin via long-rang transport, and caused the elevated FS concentrations (Chen et al., 2015;Tao et al., 2013). The highest contributions from FS to $PM_{2.5}$ mass was more than 10%, appeared in spring at both sites. The majority of $PM_{2.5}$ components showed a summer minimum, which was caused by high planetary boundary layer height favoring pollutants dispersion and abundant precipitation favoring wet scavenging (Table 1). One exception was $SO_4^{2-}$, which had a minimum in spring at CD and in autumn at CQ, likely due to the enhanced photochemical reactions associated with high temperature and strong solar radiation in summer. High $O_3$ concentrations in summer also supported this seasonal trend. It is also noted that the seasonal variations of $NO_3^-$ were much larger than those of $SO_4^{2-}$ and $NH_4^+$. $SO_4^{2-}$ and $SO_2$ showed similar seasonal trends, with their concentrations 1.4-2.0 times higher in winter than in the other seasons (Table 1). In contrast, the seasonal variations of $NO_3^-$ were much larger than that of $NO_2$, e.g., while the concentrations of $NO_2$ were 1.2-1.6 times higher in winter than in the other seasons, those of $NO_3^-$ were 9.6 times higher in winter than in summer at CQ. Thus, seasonal variations of $SO_2$ and $NO_x$ emissions were comparable, but the atmospheric chemical processes caused the much larger seasonal variations in $NO_3^-$. The concentration of $NO_3^-$ could be enhanced in winter under high RH through heterogeneous aqueous processes and decreased in summer due to volatility of $NH_4NO_3$ under high temperature, which increased the seasonal differences in $NO_3^-$ concentrations between winter and summer (Pathak et al., 2009;Quan et al., 2015;Squizzato et al., 2013). In addition, thermodynamically driven behavior of $NH_4NO_3$ was another factor for the lower $NO_3^-$ concentrations in summer (Wang et al., 2016;Kuprov et al., 2014). As shown in Fig. 2, the seasonal average contributions of SNA to $PM_{2.5}$ only varied within a small range from 39.5% to 43.2% at CD, whereas in a relatively larger range from 31.0% in summer to 37.1-41.5% in the other seasons at CQ. The smaller contribution in summer at CQ was mainly due to the lower $NO_3^-$ concentrations. At

both CD and CQ, $NO_3^-$ and $NH_4^+$ showed the highest contributions in winter and the lowest ones in summer, whereas an opposite trend was found for $SO_4^{2-}$. Both OC and EC exhibited the highest concentrations in winter at CD and CQ, around 1.9-3.1 times of those in the other seasons. SOC was also the highest in winter at both sites, similarly to what observed for OC, which can be partly explained by the enhanced condensation process forming SOC under low temperature (Sahu et al., 2011;Cesari et al., 2016). In contrast, high temperature in summer favored gas-particle partitioning in the gas phase and thus limited the formation of SOC (Strader et al., 1999). The contributions of carbonaceous components generally followed the seasonal patterns of SNA, accounting for 26.7-38.8% of $PM_{2.5}$ mass. Among these, OM showed the lowest fractions in $PM_{2.5}$ in spring (21.1%) at CD and the highest value in winter (33.6%) at CQ, while the percentages of OM in other seasons were similar at both sites, around 27%. The seasonal variations of EC fractions were not obvious, with a slightly higher value in spring.

### 3.1.3 Similarities and differences between the two sites

Although none of the two sites alone can represent the whole region of the Sichuan Basin, the similarities in the characteristics of the major pollutants between the two sites should represent the regional-scale characteristics of urban-environment pollution while the differences between the two sites should reflect the sub-regional characteristics of urban pollution. A comparison between the two sites in terms of seasonal-average concentrations of major chemical components is shown in Fig. 4 and discussed in detail below. Despite the 260 km distance between the two sampling sites, a moderate similarity was observed in autumn, winter and spring on the basis of low COD values (0.15-0.18), indicating limited differences between the two urban environments in the Sichuan Basin and the similarities in major emission sources for both sites. The similar pollution patterns observed at both CD and CQ were likely to be related to the similar meteorological parameters and special topography of the basin, which is a closed lowland surrounded by high mountains on all sides (Fig. 1). The mean elevation in the basin is about 200-700 m, while the surrounded mountains are around a scope of 1000-3000 m elevation. The Tibetan Plateau lies close to the western Sichuan Basin, with an elevation above 4000 m. Such a Plateau-Basin topography forms a barrier for the dispersion of pollutants and causes air stagnation within the basin, thereby facilitating regional scale pollution events in the basin. 72-h air mass back trajectory analysis (18:00 local time) showed that air masses reaching at CD and CQ mainly originated from local areas in the basin (Fig. S1), confirming the influence of the high mountainous surrounding the basin. These results were consistent with those found in earlier studies in Chengdu and Chongqing (Tian et al., 2017;Liao et al.,

2017), which suggested that air masses had short-range trajectories and primarily originated from inside the Sichuan Basin, highlighting the impacts of the special topography on $PM_{2.5}$ pollution. A similar case has also been found elsewhere, such as in Po Valley, Italy (Ricciardelli et al., 2017).

It is worth to note that the COD values used to identify the similarities or differences of the two sites were calculated based on seasonal-average concentrations of all the components in $PM_{2.5}$. However, if focusing on individual components, several chemical species in $PM_{2.5}$ differed by up to a factor of 2.5 in their season-average concentrations between CD and CQ, e.g. OC and EC in winter and spring, and $Cl^-$ and FS in all the four seasons. In summer, the differences for several major chemical components (FS, OC, $SO_4^{2-}$, $NO_3^-$ and EC) between the two sites were larger than in the other seasons, causing a high COD value (0.33). These discrepancies were partly caused by the different atmospheric chemical processes, local sources and meteorological parameters between the two sites. Specifically, FS mostly deviated from the 1:1 straight line in all the seasons, with substantially higher concentrations at CQ than CD (Fig. 4). There was no significant difference in $NH_4^+$ concentrations between CD and CQ, but considerable differences in $SO_4^{2-}$ and $NO_3^-$ in spring and summer. $SO_2$ concentration was around 25% higher at CQ than CD in spring and summer, which partially explained the site-differences in $SO_4^2$. In contrast, $NO_2$ concentration was comparable at both sites in summer, but $NO_3^-$ concentration was 58% lower in CQ than CD. The site-differences in $NO_3^-$ concentration was caused by $NH_4NO_3$ thermodynamic equilibrium controlled by ambient temperature and RH, instead of by its gaseous precursors. The equilibrium would be shifted toward the particulate phase when ambient RH was above the deliquescence relative humidity (DRH) of $NH_4NO_3$, and the dissociation constant decreased by about one order of magnitude when RH was above 75% (Kuprov et al., 2014). DRH was calculated from temperature following Mozurkewich (1993). As shown in Table 1, the average temperature was comparable at CD and CQ during the summer period, hence leading to similar DRH values of $NH_4NO_3$, ranging from 59% to 64% with an average value of 60.7%. However, the ambient RH was substantially lower at CQ (61%) than CD (72%), causing lower $NO_3^-$ concentration at CQ. As shown in Fig. S2, 53% of the hourly data in summer having ambient RH lower than DRH at CQ, while only 19% such data at CD, which explained the different $NO_3^-$ concentrations between CD and CQ.

Fig.4 shows higher concentrations of carbonaceous component (OC and EC) at CQ than CD in all the seasons except OC in autumn and EC in winter. OC and EC mainly originate from fossil fuel combustion and biomass burning. $K^+$ is usually regarded as a tracer of biomass burning (Tao et al.,

2016). During the sampling campaign, no significant differences in $K^+$ levels were observed between CD and CQ (Table 1), suggesting that biomass burning was not be the major cause of the higher concentrations of carbonaceous component at CQ. Motorcycle traffic was likely a major source of volatile organic compounds (VOCs) in CQ since it is a famous mountain city where most people use motorcycle as daily traffic tools. The number of motorcycles was 2.0 million in Chongqing in 2014, which was much higher than those (0.7 million) in Chengdu (Chongqing and Chengdu Statistical Yearbook 2015). Moreover, Chongqing has become China's largest automobile production base, which likely also emit VOCs from the spraying process. Higher concentrations of VOCs in CQ would cause higher concentrations of secondary organic carbon via photochemical reaction under high temperature or vapor condensation under low temperature. This hypothesis is supported by the large differences in OC concentrations in winter between the two sites.

Correlation analysis may also provide an insight into the similarities/differences between the two sites over an intensive sampling period. Good correlations between the two sites were found for daily SNA, OC, EC and $K^+$ concentrations in autumn, winter and spring (Table S1). However, for $NO_3^-$, a significant correlation was identified only in autumn, likely due to the strong impact of local vehicle emissions and the subsequent atmospheric processes forming $NO_3^-$. Similarly, a moderate correlation was observed just in winter for both $Cl^-$ and FS. In summer, weak or no correlations were identified between the two sites for almost all chemical components.

**3.2 PM$_{2.5}$ formation mechanisms and geographical origins**

**3.2.1 Pollution episodes and key chemical components**

Pollution episodes during the campaign are highlighted with shaded areas in Fig. 5. These pollution periods (PP) were defined as daily PM$_{2.5}$ concentration being above the NAAQS guideline value of 75 $\mu g \ m^{-3}$. Similarly, the days with PM$_{2.5}$ concentration below 75 $\mu g \ m^{-3}$ were characterized to be clean periods (CP). A total of seven pollution episodes were identified during the campaign at each site. There were three long-lasting pollution episodes occurred simultaneously at the two sites on 23-27 October 2014, 7(8)-26 January, and 26-28 (29) April 2015. A total of 34 and 31 pollution days were counted at CD and CQ site, respectively, accounting for 30.4% and 28.6% of the entire sampling days (112 days). The number of pollution days at CD was 8, 21, 4 and 1 in autumn, winter, spring and summer, accounting for 29.6%, 75%, 14.3% and 3.4% of the total sampling days in each season, respectively, and at CQ they were 4, 19, 6 and 2, accounting for 14.8%, 67.9%, 21.4% and 6.9%. Stagnant

atmosphere and high RH were important factors causing $PM_{2.5}$ pollution events, as was found in this and earlier studies (Zheng et al., 2015b;Chen et al., 2017;Liao et al., 2017). Compared with the clean periods, the pollution periods were usually characterized by low planetary boundary layer height and weak wind speed (Table S2), which suppressed pollutants dispersion vertically and horizontally. Temperature increased during the long-lasting pollution episodes, which promoted gas-to-particle transformation, forming secondary aerosols. RH remained high (68-88%) during pollution episodes (except in spring at CQ), although not much different from clean periods, which was also conducive for aqueous-phase reactions converting gaseous pollutants into aerosols (Chen et al., 2017;Tian et al., 2017).

Looking more closely at a regional-scale long-lasting pollution episode in winter, from 8 January to 26 January 2015, the concentrations of $PM_{2.5}$ and major chemical components increased dramatically compared with clean periods (Fig. 6). $PM_{2.5}$ concentrations were more than three times higher at both sites, with the two dominant groups of components, SNA and OC, being 2.5-2.8 times higher at CD and 1.7-2.7 times higher at CQ. The enhancement of SNA and OC during pollution periods were similar at CD, but OC increased much more than SNA at CQ, indicating some different contributing factors to the high $PM_{2.5}$ pollution at the two sites. Pollutants accumulation under stagnant meteorological conditions might be a main factor at CD based on the similar magnitudes of the enhancements of $PM_{2.5}$ and its dominant components, while additional processes should have increased OC more than other components at CQ. The percentage contributions of SNA to $PM_{2.5}$ were similar during clean and pollution periods, 38-41% at CD and CQ (Fig. S3). However, the percentage contributions of OM to $PM_{2.5}$ decreased from 30.1% on clean days to 27.5% on pollution days at CD, and increased from 26.9% to 34.9% at CQ. Concentrations of the individual SNA species ($SO_4^{2-}$, $NO_3^-$ and $NH_4^+$) increased by a factor of 2.5-3.3 on pollution days compared with clean days in all the cases (Fig. 6). But the percentage contributions differed among the species as $NO_3^-$ increased and $SO_4^{2-}$ decreased on pollution days. The percentage contributions of SNA and OM to $PM_{2.5}$ discussed above were different from those found in eastern coastal China and North China Plain, where considerable increases were found for SNA and decreases for OM on pollution days than clean days (Tan et al., 2009;Wang et al., 2015a;Quan et al., 2014;Zhang et al., 2015;Zhang et al., 2016;Cheng et al., 2015). The pollution periods in eastern coastal China and North China Plain were accompanied with sharp increases of RH, which would promote aqueous-phase formation of secondary inorganic aerosols and resulted in rapid elevation of

$SO_4^{2-}$ and $NO_3^-$ concentrations (Zheng et al., 2015b;Zheng et al., 2015a;Zhao et al., 2013b;Li et al.,
2017a). In contrast, RH remained high during both clean or pollution periods in the present study,
suggesting that high RH might not be the primary cause of the dramatic increase in $PM_{2.5}$ concentrations
during the pollution period in the Sichuan Basin. Another point that need to be mentioned is that, as
shown in Fig. S1, local sources were the main contributors to the pollution episodes in the Sichuan
Basin while sources outside local regions frequently contributed to pollution episodes in eastern coastal
China and North China Plain through long/medium range transport (Gao et al., 2015;Hua et al.,
2015;Wang et al., 2015b).
**3.2.2 Transformation mechanisms of secondary aerosols**
In most cases, meteorological conditions, atmospheric chemical processes and long-range transport are
all responsible for $PM_{2.5}$ accumulation (Zheng et al., 2015b). CO is directly emitted from combustion
processes and is not very reactive. Its concentrations in the air are strongly controlled by meteorological
parameters within a relatively short period, which make it a good tracer that can be used for separating
different dominant factors contributing to pollutants accumulation (Zheng et al., 2015b;Zhang et al.,
2015;Hu et al., 2013;Liggio et al., 2016). The impact of atmospheric physical processes on other
pollutants can be revealed by scaling the concentrations of other pollutants to that of CO. For example,
$PM_{2.5}$ was enhanced by a factor of 2.7 on pollution days at both sites, but the CO-scaled $PM_{2.5}$ (the ratio
of $PM_{2.5}$ to CO concentration) only showed an enhancement of a factor of 1.6-1.8 (Fig. 7), and the latter
values were likely from the enhanced secondary aerosol formation.
As shown in Fig. 7, the CO-scaled SNA was 60-90% higher on pollution days with individual
species 40-120% higher, even though their gaseous precursor ($SO_2$ and $NO_2$, no data for $NH_3$) were
only less than 30% higher. This suggested stronger chemical transformation from gaseous precursors to
particle formation on pollution days. Sulfur oxidation ratio ($SOR = n\text{-}SO_4^{2-}/(n\text{-}SO_4^{2-}+n\text{-}SO_2)$) and
nitrogen oxidation ratio ($NOR = n\text{-}NO_3^-/(n\text{-}NO_3^-+n\text{-}NO_2)$) were defined to evaluate the degree of
secondary transformation (*n* refers to as the molar concentration) (Hu et al., 2014). NOR increased from
0.09 on clean days to 0.16 on pollution days at CD and from 0.07 to 0.14 at CQ. SOR increased only
slightly, from 0.31 to 0.35 at CD and 0.28 to 0.35 at CQ. The CO-scaled SOC increased by a factor of
2.6 on pollution days at CQ, but no significant change was found at CD. The different patterns in SOC
(or SOC/OC) than SNA (or SOR and NOR) suggested that secondary organic aerosols (SOA)
production was of less important than SNA production at CD.

$SO_4^{2-}$ is predominantly formed via homogeneous gas-phase oxidation. In this pathway, $SO_2$ is

firstly oxidized by OH radical to $SO_3$, and then to $H_2SO_4$ (Stockwell and Calvert, 1983;Blitz et al.,
2003). Apart from homogeneous reaction, particulate $SO_4^{2-}$ can also be formed through heterogeneous
reactions with dissolved $O_3$ or $H_2O_2$ under the catalysis of transition metal and in-cloud process
(Ianniello et al., 2011). $HNO_3$ is primarily produced from the reactions between $NO_2$ and OH radical
during the daytime and later combines with $NH_3$ to produce particulate $NO_3^-$ (Calvert and Stockwell,
1983). Particulate $NO_3^-$ can also be formed through heterogeneous hydrolysis of $N_2O_5$ on moist and
acidic aerosols during nighttime (Ravishankara, 1997;Brown and Stutz, 2012). Similarly, SOA is
mainly formed through photochemical oxidation of primary VOCs followed by condensation of SVOC
onto particles as well as through aqueous-phase reactions (Ervens et al., 2011). While photochemical
reactions are mostly influenced by temperature and oxidants amount, heterogeneous reactions always
depend on ambient RH. To further explore the formation mechanisms of secondary aerosols, SOR,
NOR and SOC/OC data were grouped with temperature (at 2°C interval), RH (at 5% interval) and
daytime $O_3$ concentration (at 5 $\mu g\ m^{-3}$ interval) bins (Fig. 8). An obvious increase of SOR with
increasing RH was found at both sites, but this was not the case for temperature and $O_3$ concentration,
suggesting heterogeneous processes played important roles in the formation of $SO_4^{2-}$, as was suggested
in many previous studies (Quan et al., 2015;Zheng et al., 2015a;Zhao et al., 2013b). Interestingly, SOR
exhibited a decreasing trend with increasing $O_3$ concentration at $O_3$ concentrations lower than 15 $\mu g\ m^{-3}$
and an opposite trend was found at $O_3$ concentrations above 20 $\mu g\ m^{-3}$ (Fig. 8). Additionally, high $PM_{2.5}$
concentrations were mostly associated with lower $O_3$ concentrations. This behavior might be explained
by the complicated interactions between aerosol and $O_3$. On one hand, aerosols are generally considered
as a constraining factor to $O_3$ production due to their absorption and scattering of UV radiation, which
reduce solar radiation and consequently decrease photochemical activity. On the other hand, aerosols
can provide an interface for the heterogeneous reaction, in accordance with $O_3$ consumption and
secondary aerosol formation, which would result in decreased $O_3$ concentrations and increased
secondary aerosols (Zheng et al., 2015b). It was further found that the ambient RH remained high at low
$O_3$ concentrations (Fig. S4), which was beneficial to $SO_4^{2-}$ formation through heterogeneous aqueous
processes, consistent with the observed results that high SOR value occurred at low $O_3$ concentrations.

Unlike SOR, NOR increased with both temperature and RH, suggesting the combined effects from

homogeneous and heterogeneous reactions. However, under the very high temperature and RH
conditions, NOR exhibited a decreasing trend with increasing temperature and RH. Volatilization of
$NH_4NO_3$ at high temperature should be the major cause of such a phenomenon, but it is not clear about
the cause of the decreasing trend of NOR under high RH. Pathak et al. (2009) investigated the formation
mechanism of $NO_3^-$ in ammonium-rich and ammonium-poor samples, suggesting homogeneous gas-
phase reaction became evident to form $NO_3^-$ under the former condition while heterogeneous process
dominated the $NO_3^-$ formation under the latter condition. As shown in Fig. 9, $SO_4^{2-}$ and $NO_3^-$ were
almost completely neutralized by $NH_4^+$, indicating an ammonium-rich environment during the sampling
campaign. The ammonium-rich environment was also confirmed by the molar ratios of $[NO_3^-]/[SO_4^{2-}]$
and $[NH_4^+]/[SO_4^{2-}]$. The molar ratio $[NO_3^-]/[SO_4^{2-}]$ as a function of $[NH_4^+]/[SO_4^{2-}]$ is depicted in Fig. 9,
which shows significant positive correlations ($R^2$=0.82-0.83 at the two sites). Linear regression
equations were obtained as $[NO_3^-]/[SO_4^{2-}] = 0.85[NH_4^+]/[SO_4^{2-}]-1.89$ at CD and $[NO_3^-]/[SO_4^{2-}] =$
$0.92[NH_4^+]/[SO_4^{2-}]-1.82$ at CQ. Based on these equations, the molar ratio of $[NH_4^+]/[SO_4^{2-}]$ was defined
as the threshold value separating ammonium-rich and ammonium-poor conditions when the value of
$[NO_3^-]/[SO_4^{2-}]$ was zero. In the present study, the threshold value was 2.2 and 2.0 at CD and CQ,
respectively. The molar ratio $[NH_4^+]/[SO_4^{2-}]$ was higher than the threshold value at both sites,
suggesting the prevalence of ammonium-rich condition. The near-perfect fitting between the molar
ratios of $[NO_3^-]/[SO_4^{2-}]$ and $[NH_4^+]/[SO_4^{2-}]$ further demonstrated the characteristics of $NO_3^-$ formed
through homogenous gas-phase reaction. Moreover, $NO_3^-$ showed a strong correlation with excess $NH_4^+$
with correlation coefficients of 0.98-0.99 at both sites, providing an insight into the gas-phase reactions
of ambient $NH_3$ and $HNO_3$. Using high-resolution inorganic ions data, Tian et al. (2017) demonstrated
that $NO_3^-$ was primarily formed via homogeneous reaction when RH below 75% and through
heterogeneous processes when RH was above 75%. The increases in NOR with RH at both sites also
revealed the possibility of the heterogeneous processes, although this cannot be verified directly due to
the lack of high-resolution data.
The ratio of SOC/OC decreased with increasing temperature at CD but increased at CQ when
temperature was lower than 10°C. Although SOC/OC did not correlate well with RH, an opposite trend
was also found between CD and CQ at high RH conditions. Heterogeneous reactions seemed to be
dominant in the formation of SOA at CD, whereas homogeneous reactions were prevalent at CQ.
SOC/OC showed no apparent dependency on $O_3$ concentrations at either site, indicating more complex
formation mechanisms of SOA than $SO_4^{2-}$ and $NO_3^-$.

**3.2.3 Geographical origins of high PM$_{2.5}$ pollution**

PSCF analysis was applied to investigate the potential source regions contributing to high PM$_{2.5}$ pollution. As can be seen from the PSCF maps in Fig. 10, all the pollutants including PM$_{2.5}$ and its chemical components as well as gaseous precursors had similar spatial patterns of potential source areas. Basically, all the major source areas for high pollutants concentrations were distributed within the basin. Long-range transports as seen in North Plain and eastern coastal regions were not observed at CD and CQ (Zhao et al., 2015;Zhang et al., 2013). At CD, the major source areas in winter included the areas of the northeastern, southeastern and southern Chengdu and in some areas of eastern Chongqing. A similar spatial distribution of PM$_{2.5}$ potential sources was also found by Liao et al. (2017) through PSCF analysis in winter 2013, in which high PM$_{2.5}$ concentrations were mostly associated with sources broadly located in the southeast of the basin, covering Neijiang, Zigong, Yibin, Luzhou and east part of Chongqing. At CQ, the northeast area of Chongqing was identified as strong sources, where a number of industries were located, such as Changshou chemical industrial ozone. Overall, PM$_{2.5}$ pollution at CQ was characterized by significant local contribution from major sources located in or nearby Chongqing. In contrast, regional transport in Sichuan Basin from southeast, south and southwest of Chengdu had a major impact on PM$_{2.5}$ pollution at CD.

**4 Conclusions**

Chemically-resolved PM$_{2.5}$ data collected during four seasons at two urban sites in Sichuan basin, southwest China were analyzed in the present study. On about 30% of the days, daily PM$_{2.5}$ exceeded the national air quality standard, with annual mean concentrations of $67.0 \pm 43.4$ and $70.9 \pm 41.4$ μg m$^{-3}$ at CD and CQ, respectively. SO$_4^{2-}$, NO$_3^-$, NH$_4^+$, OM, EC and FS were the major chemical components of PM$_{2.5}$, accounting for 16.8%, 13.6%, 10.8%, 26.1%, 5.4%, and 5.7% of PM$_{2.5}$ at CD, and 17.2%, 10.9%, 9.2%, 29.6%, 6.4%, and 9.5% at CQ, on annual average, respectively. The concurrent occurrences of heavy pollution events at the two sites and similarities in pollutants characteristics between the two sites was mainly caused by the surrounding mountainous topography under typical stagnant meteorological conditions. Such a finding was also supported by back trajectory analysis which showed that air masses reaching at both sites were originated within the basin and only traveled for short distances on heavy polluted days. Differences between the two sites with regards to several major chemical components provided evidences of sub-regional characteristics of emission sources and chemical transformation processes under different meteorological conditions. For example, an additional source factor from

motorcycle traffic was identified for VOCs emission in Chongqing, which led to higher OC
concentrations, and lower relative humidity in Chongqing caused lower $NO_3^-$ concentration in this city
despite similar levels of its gaseous precursors in the two cities. The present study also identified different
driving mechanisms for the $PM_{2.5}$ pollution episodes in the Sichuan Basin than in the other regions of
China. For example, sharply increased relative humidity was thought to be one of the main factors causing
high inorganic aerosol concentrations during the pollution periods in eastern coastal China and North
China Plain, while in the Sichuan Basin the special topography and meteorological conditions are driving
forces for such events considering relative humidity was high throughout the year and did not differ much
between pollution and clean periods. However, on annual basis heterogeneous reactions might be more
important in this basin than in the other regions of China due to the consistent high humidity conditions,
as revealed in the case of $SO_4^{2-}$ formation in the present study. Future studies should use high-resolution
data to verify the findings related to chemical transformation paths proposed here. More importantly, a
detailed emission inventory of aerosol particles and related gaseous precursors in the basin should be
developed promptly, which is needed for further investigating $PM_{2.5}$ formation mechanisms and for
making future emission control policies. Source-receptor analysis using monitored chemical-resolved
$PM_{2.5}$ data should be conducted to verify such emission inventories.

*Competing interests*. The authors declare that they have no conflict of interest.

*Acknowledgements*. This work was supported by the National Natural Science Foundation of China (No.
41405027, 41375123, and 41403089), the "Strategic Priority Research Program" of the Chinese Academy
of Sciences (No. KJZD-EW-TZ-G06), the West Action Plan of the Chinese Academy of Science (No.
KZCX2-XB3-14), and Chongqing Science and Technology Commission (No. cstc2014yykfC20003,
cstckjcxljrc13). We are grateful to Yumeng Zhu and Jun Wang for sample collection.

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

Table 1 Meteorological parameters, annual and seasonal mean concentrations of $PM_{2.5}$, gaseous pollutants and major chemical components at CD and CQ during 2014-2015.

| | CD | | | | | CQ | | | | |
|---|---|---|---|---|---|---|---|---|---|---|
| | Autumn | Winter | Spring | Summer | Annual | Autumn | Winter | Spring | Summer | Annual |
| Meteorological parameters | | | | | | | | | | |
| T (°C) | 15.8±2.9 | 9.3±2.5 | 20.4±4.4 | 28.3±2.9 | 18.5±7.7 | 16.0±3.2 | 10.0±2.3 | 20.5±4.5 | 28.4±3.4 | 18.8±7.6 |
| P (hPa) | 960±3.8 | 963±4.7 | 954±7.9 | 946±2.1 | 955±8.1 | 981±4.2 | 984±5.2 | 974±8.4 | 963±2.2 | 975±7.7 |
| RH (%) | 81.9±9.0 | 80.9±6.8 | 70.5±8.6 | 72.2±11.3 | 76.3±10.3 | 76.1±5.7 | 68.7±8.8 | 60.7±13.6 | 61.0±13.3 | 66.5±12.5 |
| SR (w m$^{-2}$) | 54.9±40.3 | 37.8±27.2 | 128.9±65.0 | 123.6±94.2 | 67.2±56.7 | na. | na. | na. | na. | na. |
| WS (m s$^{-1}$) | 0.5±0.2 | 0.4±0.3 | 0.7±0.4 | 0.5±0.2 | 0.5±0.3 | 0.7±0.2 | 0.7±0.3 | 1.0±0.4 | 0.7±0.3 | 0.8±0.3 |
| Precipitation (mm) | 76.3 | 18.3 | 56.6 | 247.8 | na. | 73.3 | 22.0 | 104.6 | 206.3 | na. |
| PBLH$_{max}$ (m) | 890±305 | 852±273 | 1296±491 | 1422±529 | 1119±481 | 821±252 | 928±260 | 1310±491 | 1329±505 | 1101±453 |
| Concentrations of gaseous pollutants (µg m$^{-3}$) | | | | | | | | | | |
| O$_3$ | 19.3±10.5 | 11.9±7.6 | 69.3±22.9 | 90.5±33.3 | 48.2±39.6 | 13.3±8.9 | 12.5±7.7 | 56.3±23.5 | 42.8±17.2 | 31.5±24.5 |
| SO$_2$ | 15.8±7.0 | 21.5±9.5 | 11.2±6.3 | 11.3±4.7 | 14.9±3.7 | 16.4±4.6 | 23.3±9.2 | 13.9±5.3 | 14.4±5.4 | 17.0±7.3 |
| NO$_2$ | 60.2±18.7 | 75.3±24.5 | 51.8±26.8 | 54.2±9.4 | 60.4±22.5 | 66.5±15.0 | 81.3±19.8 | 50.8±16.7 | 51.7±20.8 | 62.4±22.0 |

Concentrations of $PM_{2.5}$ and chemical compositions (µg m$^{-3}$)

| | | | | | | | | | | |
|---|---|---|---|---|---|---|---|---|---|---|
| PM$_{2.5}$ | 62.1±38.4 | 113.5±47.8 | 48.0±25.2 | 45.1±15.2 | 67.0±43.4 | 56.3±23.6 | 115.1±53.9 | 58.3±24.6 | 54.2±16.2 | 70.9±41.4 |
| SO$_4^{2-}$ | 10.5±6.5 | 16.4±7.1 | 8.3±5.9 | 9.7±4.7 | 11.2±6.8 | 9.9±4.7 | 17.5±7.4 | 10.4±6.5 | 11.1±5.7 | 12.2±6.8 |
| NO$_3^-$ | 9.3±7.4 | 17.5±8.8 | 5.9±3.6 | 3.9±2.2 | 9.1±8.0 | 7.8±3.8 | 15.8±9.5 | 5.9±5.0 | 1.6±1.3 | 7.7±7.6 |
| NH$_4^+$ | 6.9±4.8 | 12.7±5.4 | 5.1±3.2 | 4.2±1.9 | 7.2±5.2 | 5.7±2.8 | 11.3±5.2 | 5.2±3.0 | 4.0±2.1 | 6.6±4.4 |
| Cl$^-$ | 1.9±1.2 | 3.4±1.9 | 0.6±0.4 | 0.2±0.2 | 1.5±1.7 | 0.8±0.4 | 1.6±1.2 | 0.5±0.5 | 0.04±0.03 | 0.7±0.9 |
| K$^+$ | 0.6±0.4 | 1.2±0.6 | 0.6±0.5 | 0.5±0.2 | 0.7±0.5 | 0.5±0.2 | 1.2±0.7 | 0.5±0.2 | 0.3±0.1 | 0.6±0.5 |
| OC | 10.4±6.1 | 19.7±8.4 | 6.3±3.7 | 7.4±1.5 | 10.9±7.6 | 9.7±4.7 | 24.2±13.6 | 10.0±5.1 | 8.5±3.4 | 13.1±10.0 |
| EC | 3.0±2.1 | 6.3±3.0 | 2.7±2.3 | 2.5±0.7 | 3.6±2.7 | 3.8±1.7 | 5.9±3.2 | 4.7 ±3.0 | 3.7±1.5 | 4.5±2.6 |
| FS | 3.2±1.6 | 4.5±2.0 | 4.8±3.0 | 2.7±1.5 | 3.8±2.2 | 5.0±2.8 | 6.3±3.3 | 9.1±7.6 | 6.5±4.0 | 6.7±5.0 |

na. means no data.

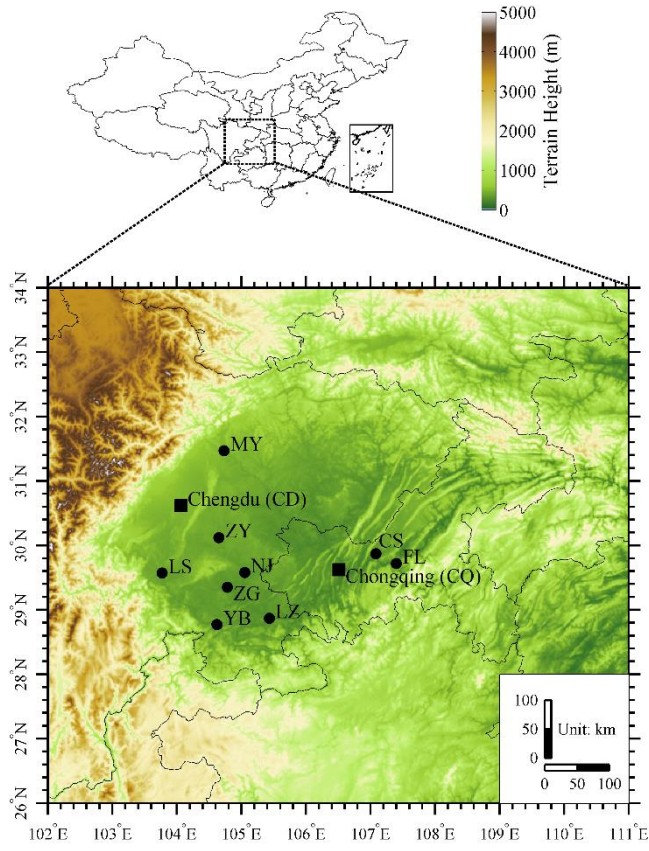

Figure 1. Locations of the sampling sites in Chengdu (CD) and Chongqing (CQ) and major cities in the Sichuan Basin. MY, Mianyang; ZY, Ziyang; LS, Leshan; NJ, Neijiang; ZG, Zigong; YB, Yibin; LZ, Luzhou; CS, Changshou; FL, Fuling.

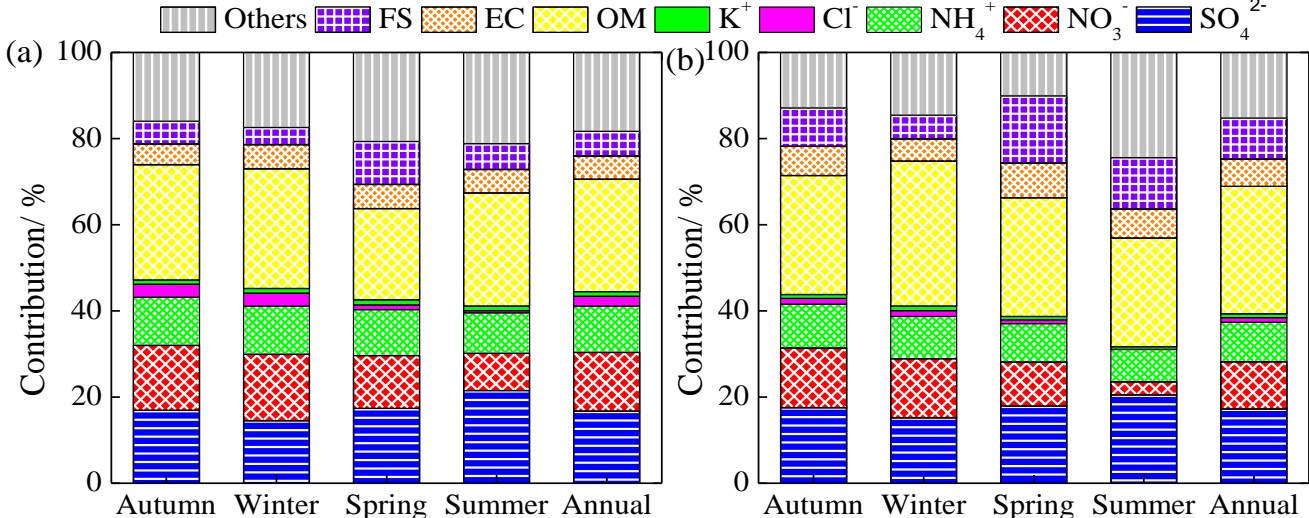

Figure 2. Seasonal and annual contributions of individual chemical components to PM₂.₅ at CD (a) and CQ (b).

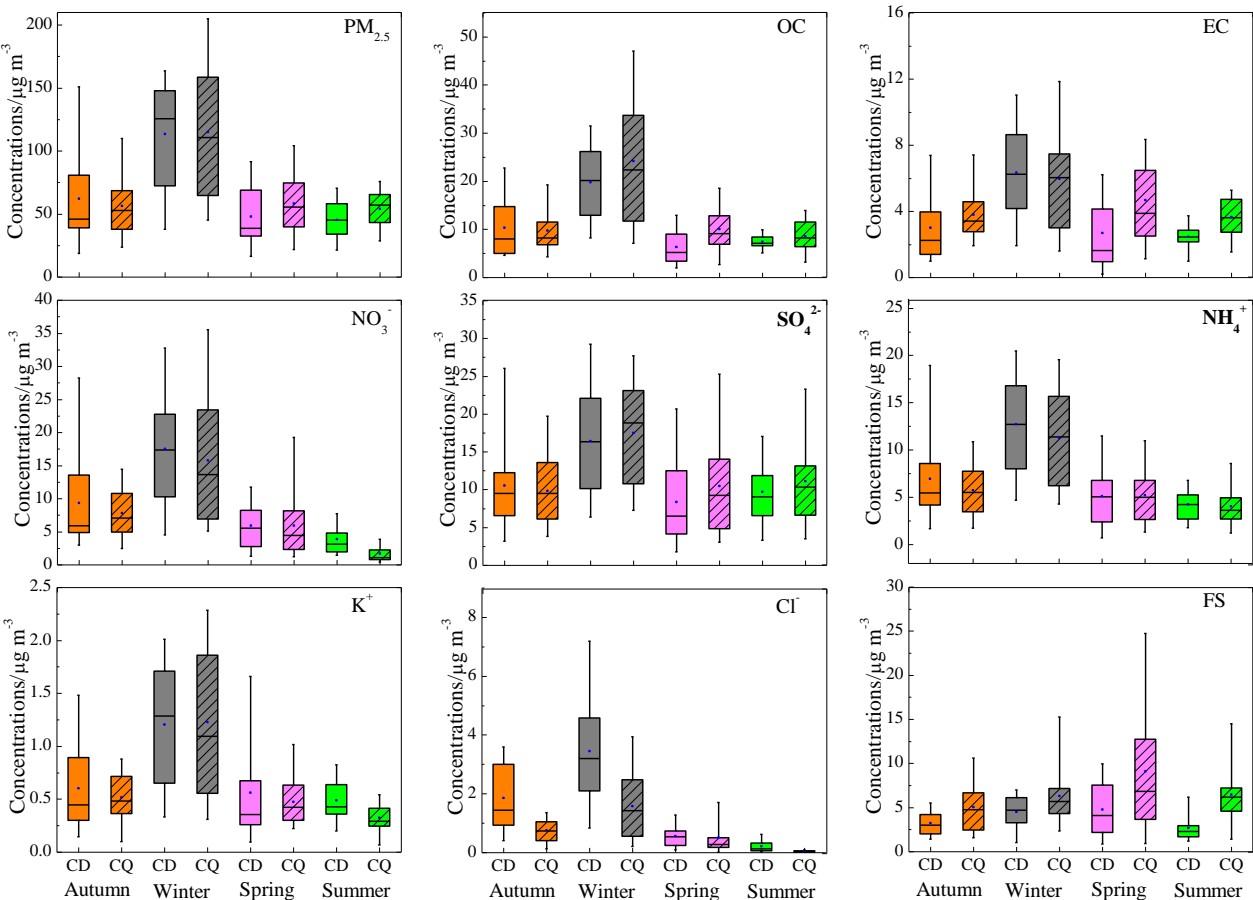

Figure 3. Seasonal distributions of PM2.5 and its major chemical components. Shown in each sub-figure are mean (dot symbol), median (horizontal line), the central 50% data ($25^{th}$ -$75^{th}$ percentiles, box), and the central 90% data ($5^{th}$-$95^{th}$ percentile, whisker)

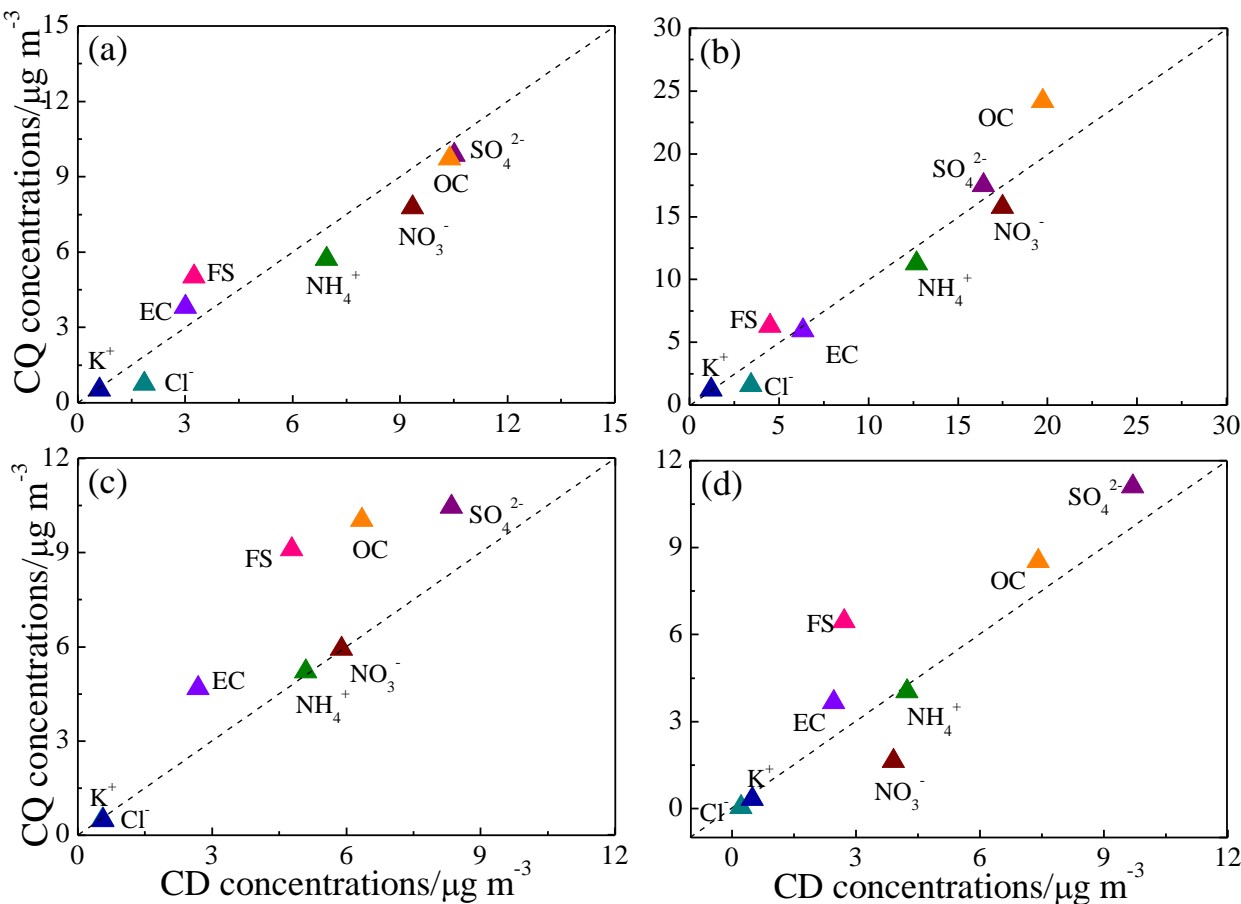

Figure 4. Seasonal mean concentrations of major components in autumn (a), winter (b), spring (c), and summer (d) at CD and CQ sites.

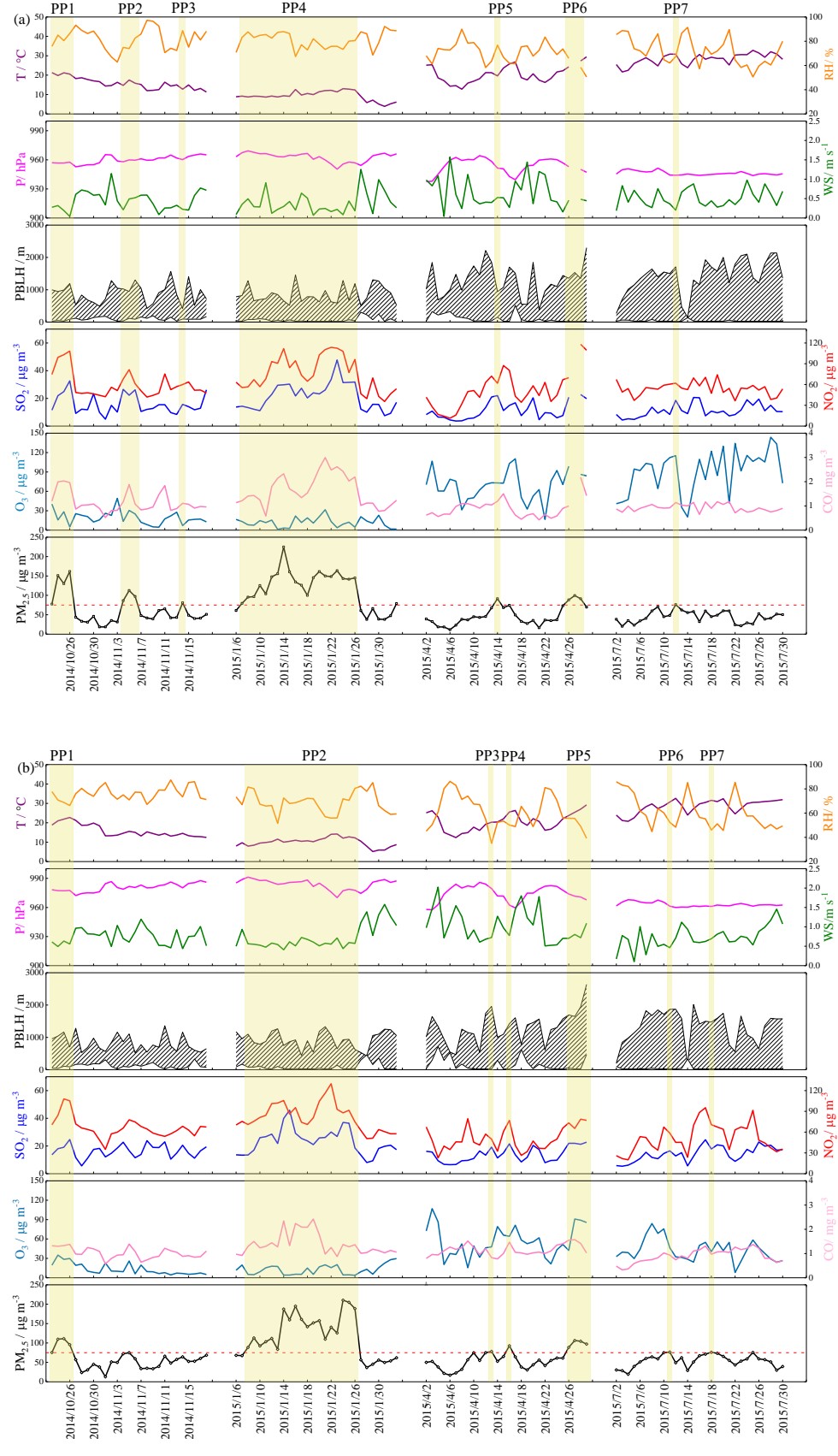

Figure 5 Temporal variations of meteorological parameters, gaseous pollutants and PM$_{2.5}$ during the campaign at CD (a) and CQ (b). Pollution episodes are highlighted by shaded areas.

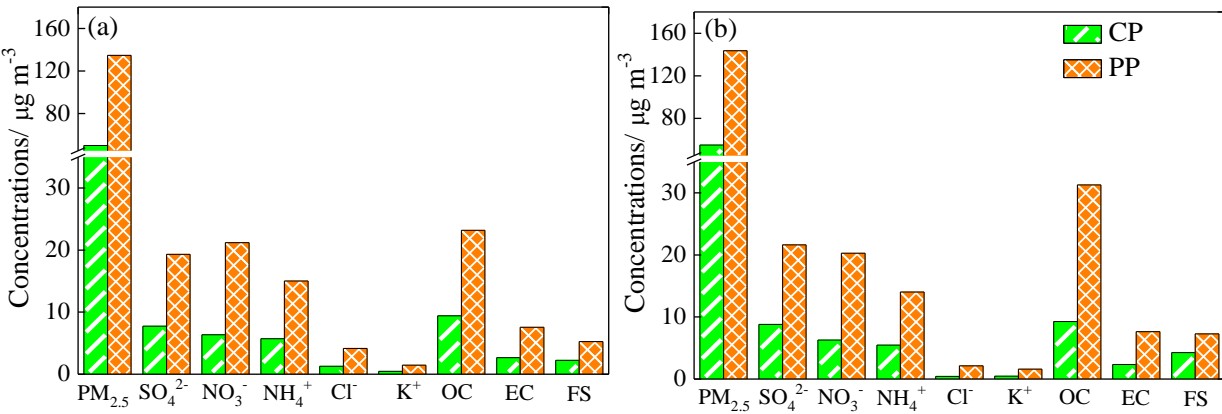

Figure 6. PM2.5 and major chemical components during clean periods (CP) and pollution periods (PP) in winter at CD (a) and CQ (b). At CD: CP, 6 January and 27 January-2 February 2015; PP, 7-26 January 2015. At CQ: CP, 6-7 January and 27 January-2 February 2015; PP, 8-26 January 2015.

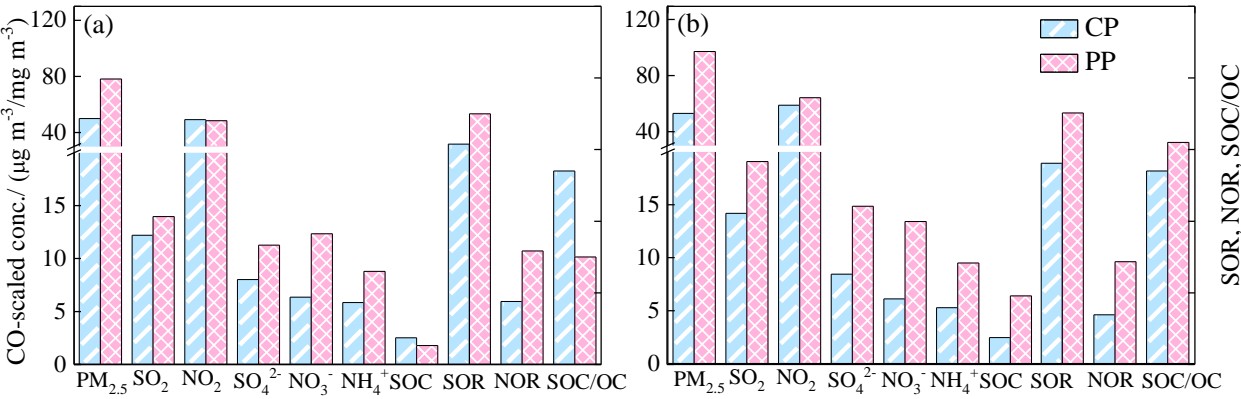

Figure 7. CO-scaled concentrations of various pollutants and the values of SOR, NOR, and SOC/OC in winter at CD (a) and CQ (b). CP and PP is the same period as Figure 6.

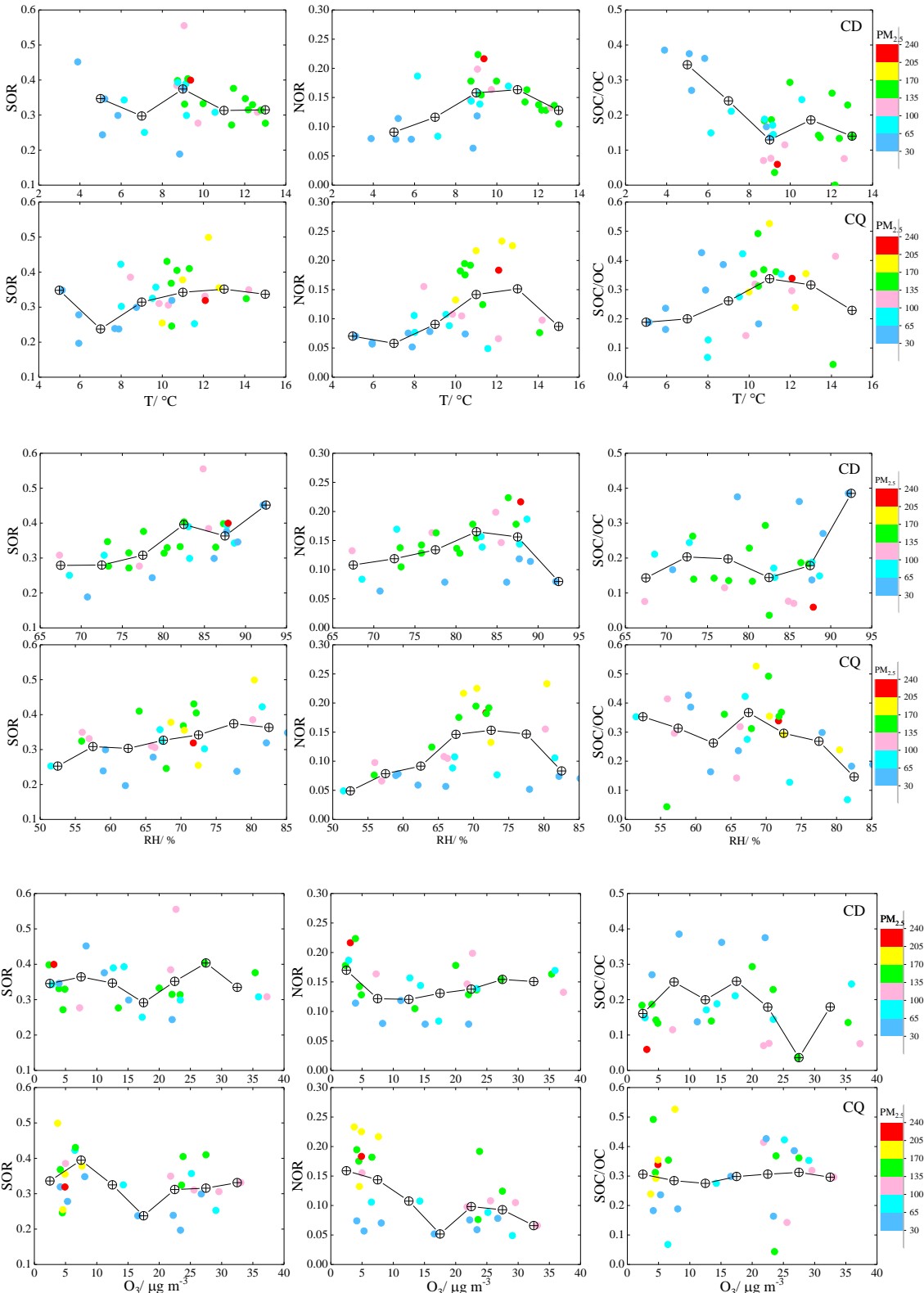

Figure 8. Correlations of SOR, NOR and SOC/OC against temperature (upper), RH (middle) and $O_3$ concentration (bottom) in winter at CD and CQ.

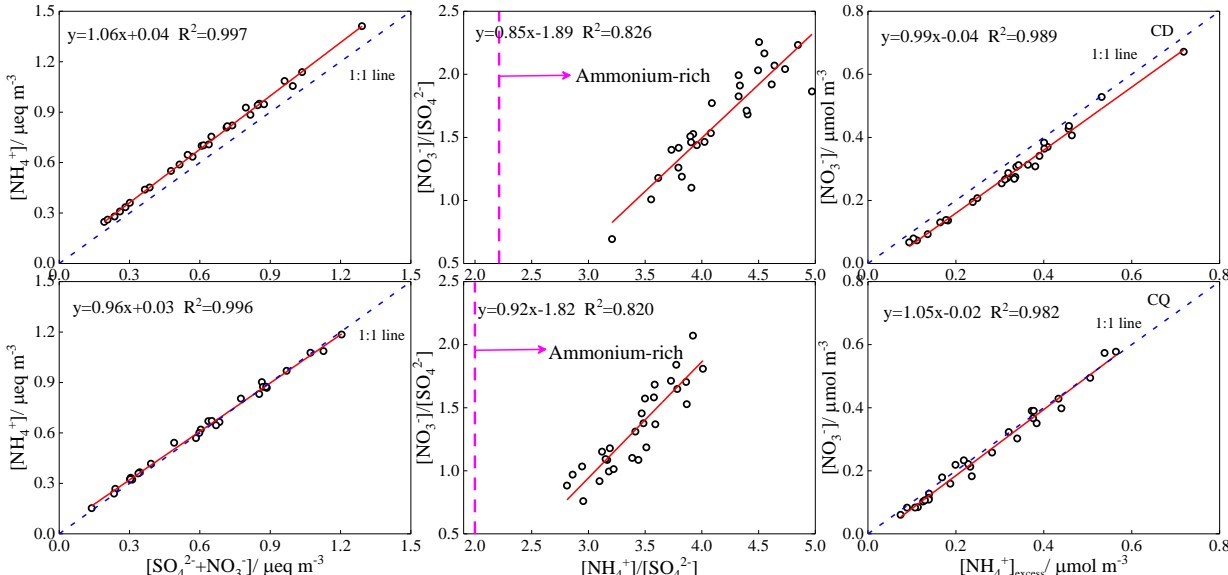

Figure 9. $NH_4^+$ concentration as a function of the sum of $SO_4^{2-}$ and $NO_3^-$ in equivalent concentrations (left column), molar ratio $NO_3^-/SO_4^{2-}$ as a function of $NH_4^+/SO_4^{2-}$ (middle column), and $NO_3^-$ concentration as a function of $NH_4^+_{excess}$ (right column) at CD (upper row) and CQ (lower row).

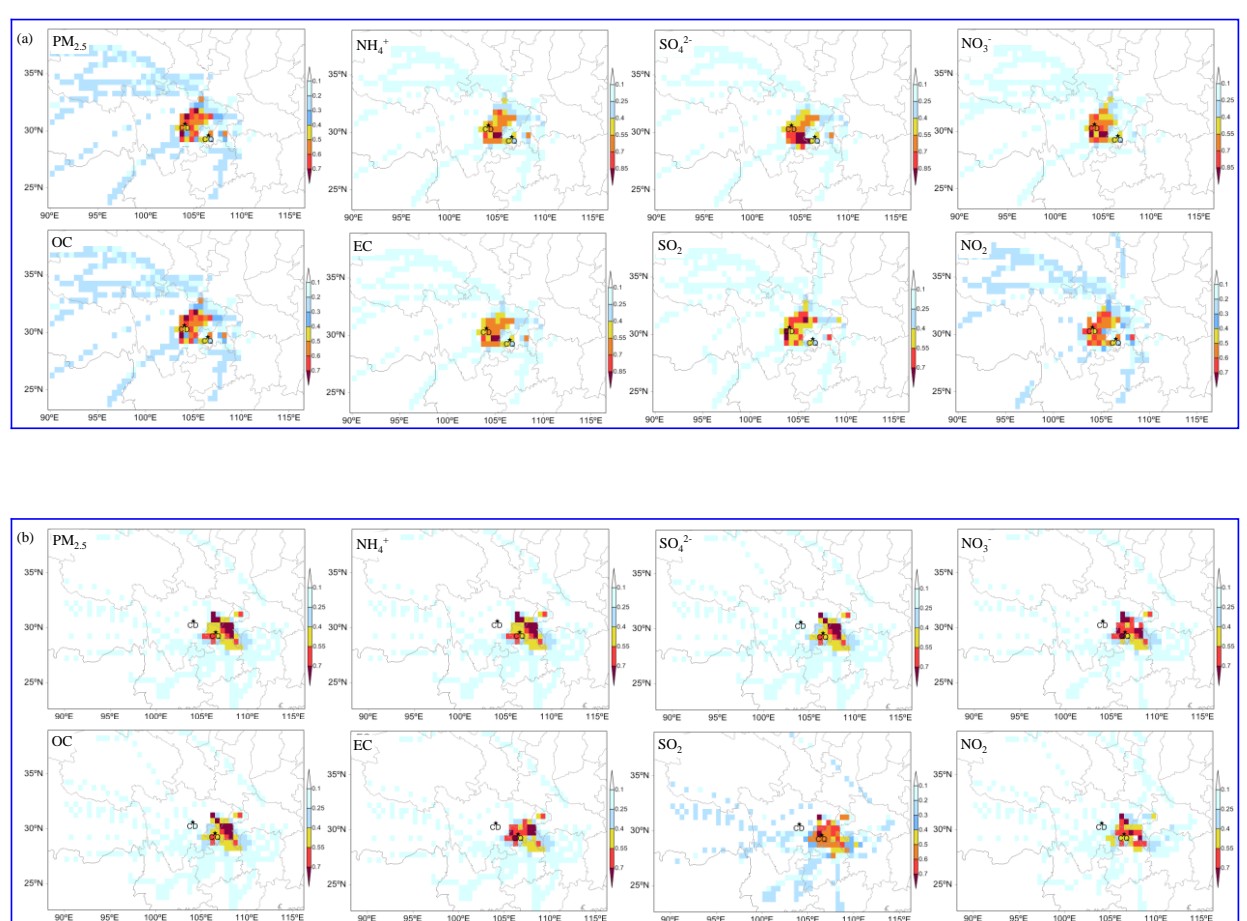

Figure 10. PSCF distribution of PM2.5, its chemical components, and gaseous precursors in winter at CD (a) and CQ (b).