# Peer review of "Seasonal characteristics, formation mechanisms and source origins of $PM_{2.5}$ in two megacities in Sichuan Basin, China"

_Atmospheric Chemistry and Physics, 2017_

## Referee Comment (RC1) · Anonymous Referee #2 · 2 May 2017

General Comments:

This manuscript elucidates the seasonal variations, the formation mechanism, and the sources of $PM_{2.5}$ in two megacities in Sichuan Basin. The concentrations of major chemical components of $PM_{2.5}$ in different seasons are investigated. The chemical characteristics of clean days and polluted days are presented to study the formation of key chemical species as well as transportation pathway of secondary aerosols. In general, the research results provide meaningful information on both formation mechanism and environmental control strategies of $PM_{2.5}$. However, there are still some key issues which need to be addressed before possible publication.

1. The factors contributing to the different temporal patterns of sulfate and nitrate should be further discussed. Specific heterogeneous reaction which may play important roles in polluted days and its major contributing components can be clearly pointed out.

2. The discussion on $PM_{2.5}$ formation process can be combined with the analysis on the variation of gaseous precursors, including $SO_2$ and $NO_2$.

3. More discussion on geographical sources of different chemical components of $PM_{2.5}$ is recommended. I also suggest the analysis on the different topography of these two megacities, which will help to better explain the impact of local emission and regional transportation.

Besides, I have some specific comments on the manuscript as follows:

1. Section 2.5: I'd like to recommend adding detailed equations of PSCF analysis for better understanding.

2. Line 170-171: Citation format error: "Tao et al. (2013, 2014)" should be corrected into "(Tao et al., 2013, 2014)".

3. Line 244: The authors claimed the concentration of $NO_3^-$ decreased on the polluted days in the warm season of CQ. But in Figure 6(d), the concentration of $NO_3^-$ is higher in the polluted days. There seems to be contradictory.

4. Section 3.4.2: The authors applied CO-scaled $PM_{2.5}$ and major components to isolate the impact of meteorological conditions. Specific scaling approach or related references should be provided.

5. Section 3.4.3: I'd like to recommend adding a graph containing RH levels and $NO_3^-$ concentration between the two sites. Also, a correlation between $[NO_3^-]/[SO_4^{2-}]$ and and $[NH_4^+]/[SO_4^{2-}]$ is suggested to investigate the difference of $NO_3^-$ formation between CQ and CD.

6. Figure 3: The black dots which indicate the average values should be stated in the figure caption.

---

## Referee Comment (RC2) · Anonymous Referee #3 · 18 Jul 2017

The manuscript is based on the observation conducted in four selected months in two cities in the Sichuan Basin, China. It represents the results of PM2.5 and the chemical components. The seasonal variations are shown and the difference in terms of the formation mechanisms and geographical influence between the two cities is discussed. The content of this manuscript fits the scope of ACP and the data is interesting to be studied. However, I found this manuscript is only a report of the results in a rarely investigated region in China but without in-depth analysis. No novel point has been raised and discussed in this manuscript. I would not recommend it to be published in ACP in the current stage.

[Figure]

General comments: 1. The sampling campaign in four selected months may not be enough to provide sufficient data to answer the questions (objectives) which are supposed to be studied in this work. The two sampling sites seems not ideal to understand the characteristics of PM2.5 in two basined cities with typical geographical features. Regional sites without direct emissions are better in my opinion. In order to discover and reveal the formation mechanisms of secondary aerosols, more data and analysis are necessary. 2. The data are not well presented in this manuscript. The readers can hardly find the sufficient information to know and understand the results. For example, how many samplers were collected in the campaign? How many samples were taken and how about the variations of data in clear days, moderate polluted days and heavy polluted days? Were there some different pollution episodes? 3. The analysis and discussion are superficial and full of speculation. No solid evidence can be provided to support the conclusions, which makes the significance and implication ungrounded. For example, to support their hypothesis, the diurnal variations of monitored gases are presented and discussed. However, the data of PM2.5 and their chemical components are on daily basis, which weaken the analysis and leads to vague conclusions.

More specific comments are shown as follows: Specific comments: 1. As I suggested above, are the two sampling sites and the data representative for this investigation on the characteristics of aerosol in the two basined cities? Obviously they are both highly affected by the traffic emissions which may bias the analysis. The topography of the two sites and the influence should be discussed. 2. Line 78: Please provide the details of the sampler. 2. Three samplers were used in this campaign. The comparison of the three samplers should be provided to show the accuracy and consistency of the data. 3. How many samples were collected? How the blank filters (lab blank and field blank) were collected? 4. Line 111-113: There were only 5 elements detected by XRF. Normally I would expect more elements could be measured by the XRF technique. Why? 5. Line 121: Please provide the details of the weather station. 6. Line 178-179: The authors pointed out that higher sulfate concentrations were found in summer. In Table 1, I found that lower sulfate average was in summer than that in winter. Please check

the data. 7. Line 178-185: The discussion on sulfate, nitrate, chloride and potassium seems superficial and arbitrary. The analysis should be based on the data from this campaign and be made with in-depth study instead of guesses. 8. Line 188-190: The high SOC content was observed in winter. In this work, the estimation of SOC mainly depends on the seasonal minimum of OC/EC. However, it should not be surprise to see high OC in winter because organic aerosols may not necessarily be only formed by secondary reaction but also by direct emissions (e.g. biomass burning). 9. Section. 3.3 discusses the difference of data between the two sites. As it known to all, the difference can be due to many possible factors (emissions, atmospheric reactivity, meteorological conditions, the surrounding terrains). It is really hard to synthesize significant information from the comparison. Therefore, more in-depth studies are necessary. 10. Line 227-238: More information should be provided for the pollution episodes. For example, how many polluted days and in which seasons were captured? How many pollution episodes were observed? 11. Line 254-256: The distinct characteristics in the urban area in the Sichuan Basin should be further investigated and discussed. How may the topography and meteorological conditions influence on the characteristics? 12. Line 271-272: "Both CO and EC concentrations increased on polluted days, suggesting the important role the meteorological condition played on PM2.5 accumulation."Why? I cannot see any link. The occurrence of CO and EC in the troposphere should be influenced by the emissions, removal mechanisms and other factors (including meteorological conditions but not exclusively). 13. Line 274-275: "CO can be considered as a reference pollutant species whose temporal variations were mainly from the impact of meteorological conditions."Why? See the comment 11. Also, I think the CO-scaling method should be further explained with more details and with references. 14. Section 3.4.2: The diurnal trends of monitored gases could not give any solid evidence to support their hypothesis. In this case, especially when the formation mechanisms of secondary aerosols are discussed, high resolution data are necessary. We should not rely on the unsolid speculation.

---

## Author Comment (AC2) · 12 Sep 2017

We greatly appreciate the helpful comments from the reviewer, which have helped us improve the paper. We have addressed all of the comments carefully, as detailed below. Our responses start with "R:".

The manuscript is based on the observation conducted in four selected months in two cities in the Sichuan Basin, China. It represents the results of PM2.5 and the chemical components. The seasonal variations are shown and the difference in terms of the formation mechanisms and geographical influence between the two cities is discussed.

[Figure]

The content of this manuscript fits the scope of ACP and the data is interesting to be studied. However, I found this manuscript is only a report of the results in a rarely investigated region in China but without in-depth analysis. No novel point has been raised and discussed in this manuscript. I would not recommend it to be published in ACP in the current stage.

R: Based on this and another reviewer's critical comments, we have added many in-depth analyses in the revised manuscript. These include: (1) providing gaseous precursors and meteorological parameters data to explain the seasonal variation trends of PM2.5 and its chemical components; (2) conducting air mass back trajectory analyses to illustrate the influence of topography on PM2.5 pollution in Sichuan Basin; (3) adding deliquescence relative humidity (DRH) analysis for the summer season to explain the different NO3- concentrations at the two sites, and (4) comparing the characteristics of PM2.5 pollution episodes in Sichuan Basin with those in the other regions of China.

The quality of the paper has been substantially improved, as demonstrated by the many novel findings. A few major ones are detailed below.

(1) The study identified different driving mechanisms for the polluted PM2.5 episodes in the Sichuan Basin than in the other regions of China. For example, sharply increased relative humidity (RH) was thought to be one of the main factors causing high inorganic aerosol concentrations during the polluted periods in eastern coastal China and North China Plain where mostly with flat terrain (Li et al., 2017; Zhao et al., 2013; Zheng et al., 2015a; Zheng et al., 2015b, Gao et al., 2015; Hua et al., 2015; Wang et al., 2015). In contrast, RH did not differ much between polluted and clear periods in Sichuan Basin. Instead, the special topography and meteorological conditions in this region resulted in the polluted PM2.5 levels, and different local topography between CD and CQ further added different pollution characteristics between the two sites. Note that Sichuan Basin is completely surrounded by high mountains and constantly characterized by low wind speeds. RH in the region is high throughout the year, which is conducive for heterogeneous reactions forming inorganic aerosols.

[Figure]

(2) The study identified the sub-regional characteristics of emission sources. An additional VOCs emission source was identified for CQ than CD based on higher OC concentrations at CQ. This additional source was attributed to motorcycle traffic in CQ since it is a famous mountain city where most people use motorcycle as daily traffic tools. According to the Chongqing Statistical Yearbook 2015, the number of motorcycles was 2.0 million (among the total of 2.3 million motor vehicles) in 2014, which was much higher than those (0.7 million) in Chengdu (Chengdu Statistical Yearbook 2015).

(3) The study identified sub-regional characteristics of inorganic aerosols. Although the whole Sichuan Basin (as the regional scale) was characterized by special topography (surrounded by mountains) and hot and humid air, there were sub-regional differences within the basin, as contrasted between the two largest cities (CD and CQ) in this region. Thus, different aerosol characteristics were found at the two sites. For example, the lower $NO_3^-$ at CQ than at CD was identified to be caused by the lower RH based on the deliquescence relative humidity (DRH) of $NH_4NO_3$ (Mozurkewich, 1993).

With these additoanl in-depth analyses and inovative results, we hope the reviewer will find ithe paper meets the standard of the jorunal.

General comments:

1. The sampling campaign in four selected months may not be enough to provide sufficient data to answer the questions (objectives) which are supposed to be studied in this work. The two sampling sites seems not ideal to understand the characteristics of PM2.5 in two basined cities with typical geographical features. Regional sites without direct emissions are better in my opinion. In order to discover and reveal the formation mechanisms of secondary aerosols, more data and analysis are necessary.

R: Based on existing literature, the four month data observed in four typical seasons should be enough for exploring the seasonal and annual patterns of aerosol pollution levels and for exploring potential formation mechanisms, as has been demonstrated in studies for other regions of China as well as for other countries (Paraskevopoulou et

al., 2015; Pietrogrande et al., 2016; Squizzato and Masiol, 2015; Ming et al., 2017; Tao et al., 2014; Wang et al., 2016; Wang et al., 2015; Zhao et al., 2013).

We agree with the reviewer that a traditional regional background site would choose a rural site far away from urban areas/local emissions in order to study the background pollution levels. However, this is not the focus of our study. Emission control policies aim to reduce PM2.5 pollution for populated regions for human health concerns. In this sense, urban areas, especially in megacities, are the major concerns. Chengdu and Chongqing are the two biggest cities in Sichuan Basin. Thus, emission sources, transportation and chemical transformation of atmospheric aerosols observed in these cities should be investigated thoroughly. The two monitoring sites selected in this study should represent the typical urban environment in their respective cities (Chen et al., 2017; Tao et al., 2014). Although none of the two sites alone would represent the whole region of the Sichuan basin, the similarities between the two sites should represent the reginal-scale characteristics of urban-environment pollution while the differences between the two sites should reflect the sub-regional characteristics of urban pollution. This reviewer seemed to have a different view of the regional-scale phenomenon than what we had in mind, which we understand, was due to the different considerations. The selection of our sites perfectly served the goals of our study.

Although the current data set is not very big, it is enough to provide many in-depth analyses, as we have done in the revised paper after incorporating both reviewers' recommendations (with more details in our responses to the specific comments below).

2. The data are not well presented in this manuscript. The readers can hardly find the sufficient information to know and understand the results. For example, how many samplers were collected in the campaign? How many samples were taken and how about the variations of data in clear days, moderate polluted days and heavy polluted days? Were there some different pollution episodes?

R: We have added more details about the sampling information in the revised

manuscript. A total of 112 samples were collected on daily basis at each site, which were 27, 28, 28 and 29 days in autumn, winter, spring and summer, respectively. To view the pollution episodes clearly, we have added a figure describing the temporal variations of daily PM2.5 concentrations, gaseous precursors and meteorological parameters during the entire study periods. The number of polluted days was 8, 21, 4 and 1 in autumn, winter, spring and summer (total 34 days), respectively, at CD, and was 4, 19, 6 and 2 (total 31 days) at CQ.

3. The analysis and discussion are superficial and full of speculation. No solid evidence can be provided to support the conclusions, which makes the significance and implication ungrounded. For example, to support their hypothesis, the diurnal variations of monitored gases are presented and discussed. However, the data of PM2.5 and their chemical components are on daily basis, which weaken the analysis and leads to vague conclusions.

R: We agree that it is ideal to have hourly aerosol data for more detailed analyses, which unfortunately could not be obtained in this study due to instruments limitations. However, in-depth analyses can be done using daily data as demonstrated in earlier studies as well as in our revised version of this manuscript (more details below in our responses to specific comments).

More specific comments are shown as follows:

1. As I suggested above, are the two sampling sites and the data representative for this investigation on the characteristics of aerosol in the two basined cities? Obviously they are both highly affected by the traffic emissions which may bias the analysis. The topography of the two sites and the influence should be discussed.

R: See our responses to point 1 of general comments above regarding regional representative of the sites. We have added discussion on topography related impacts in the revised paper. Briefly, CD is located in the west while CQ on the eastern margin of Sichuan Basin. The basin is surrounded by mountains in all directions, which forms

a barrier for pollutants dispersion and thus causes frequent stagnant air in the basin. This results in regional-scale pollution episodes in Sichuan Basin. Air mass back trajectory analysis showed that air masses reaching CD and CQ only traveled for short distances and primarily within the basin. Such a phenomenon highlighted the impacts of the special topography on PM2.5 pollution.

2. Line 78: Please provide the details of the sampler. Three samplers were used in this campaign. The comparison of the three samplers should be provided to show the accuracy and consistency of the data.

R: The samplers used in this study are described in the manuscript. At CD site, PM2.5 sampling was carried out using a versatile air pollutant sampler (URG Corp., URG-3000K, North Carolina, USA), which has three channels. One channel was used to load PM2.5 sample on Teflon filter for mass and trace elements anlysis and the other one was equipped with quatz filter for water-soluble inorganic ions and carbonaceous components analysis. At the CQ site, a low-volume aerosol sampler (BGI Corp., frmOmni, USA) operating at a flow rate of 5 L min-1 was used to collect PM2.5 samples on Teflon filter, and another sampler (Thermo Scientific Corp. Partisol 2000i, USA) with a flow rate of 16.7 L min-1 was used to collect PM2.5 samples on quartz filter. Sampling on two parallel channel at CD site and the simultaneous sampling on two instruments at CQ site allowed the contemporary chemical determination of the loading PM2.5.

The three samplers used in this study were commercial instruments and widely used in PM2.5 sampling. We examined the flow rate of each sampler before and after sampling carfully to ensure the quality of sampling.

3. How many samples were collected? How the blank filters (lab blank and field blank) were collected?

R: See our responses to point 2 of general comments. Field blanks were collected every two weeks in each season, resulting 8 filed blanks at each site. In order to

check the background contamination from the laboratory, three lab blank filters in each campaign were stored in a clean Petri slides in the dark and were analyzed the same ways as the sampling filters. The detailed information has been added in the revised manuscript.

4. Line 111-113: There were only 5 elements detected by XRF. Normally I would expect more elements could be measured by the XRF technique. Why?

R: Besides the 5 crustal elements (Al, Si, Ca, Fe and Ti) used in the study, another 16 elements including Na, Mg, S, Cl, K, V, Cr, Mn, Co, Ni, Cu, Zn, Rb, Sb, Ba and Pb were also determined by the XRF method. Among those elements, Na, Mg, Cl, S and K were discussed in the form of ions, and other metal elements accounted for less than 1% of the $PM_{2.5}$. Thus, the above 16 elements were not considered in identifying the major chemical components that were responsible for the $PM_{2.5}$ pollution.

5. Line 121: Please provide the details of the weather station.

R: We have added some details of the weather station in the revised manuscript.

6. Line 178-179: The authors pointed out that higher sulfate concentrations were found in summer. In Table 1, I found that lower sulfate average was in summer than that in winter. Please check the data.

R: We have clarified the explanation. $SO_4^{2-}$ was the highest in winter, but not the lowest in summer. In contrast, many other pollutants had the lowest in summer.

7. Line 178-185: The discussion on sulfate, nitrate, chloride and potassium seems superficial and arbitrary. The analysis should be based on the data from this campaign and be made with in-depth study instead of guesses.

R: We have rewritten this section completely with additional data of meteorological parameters and gaseous precursors. See our summary responses at the top of this file.

[Figure]

8. Line 188-190: The high SOC content was observed in winter. In this work, the estimation of SOC mainly depends on the seasonal minimum of OC/EC. However, it should not be surprise to see high OC in winter because organic aerosols may not necessarily be only formed by secondary reaction but also by direct emissions (e.g. biomass burning).

R: We are aware of the limitations of approach, but think it is a practical method for estimating SOC, as has frequently been used in literature. We noted that there was no extensive coal combustion or wood burning for domestic heating in winter due to the warm climate in this region. Therefore, biomass burning should have small effects on OC concentrations.

9. Section. 3.3 discusses the difference of data between the two sites. As it known to all, the difference can be due to many possible factors (emissions, atmospheric reactivity, meteorological conditions, the surrounding terrains). It is really hard to synthesize significant information from the comparison. Therefore, more in-depth studies are necessary.

R: We have identified major factors causing the differences between the two sites through the following in-depth analyses, such as back trajectory analyses and deliquescence relative humidity analyses (see some details in our summary responses at the top of this file).

10. Line 227-238: More information should be provided for the pollution episodes. For example, how many polluted days and in which seasons were captured? How many pollution episodes were observed?

R: We have added a figure and related information in the revised paper (also see response to point 2 of general comments above).

11. Line 254-256: The distinct characteristics in the urban area in the Sichuan Basin should be further investigated and discussed. How may the topography and meteorological conditions influence on the characteristics?

R: See our summary responses at the top of this file.

12. Line 271-272: "Both CO and EC concentrations increased on polluted days, suggesting the important role the meteorological condition played on PM2.5 accumulation." Why? I cannot see any link. The occurrence of CO and EC in the troposphere should be influenced by the emissions, removal mechanisms and other factors (including meteorological conditions but not exclusively).

R: Such a conclusion was based on this hypothesis: the variations of CO were mainly controlled by meteorological factors (Zheng et al., 2015b; Zhang et al., 2015) while those of the other pollutants were by both meteorology and chemical transformation. In this study, PM2.5 and gaseous precursors increased by a factor of 1.8-3.3 during polluted periods than clean periods while CO only increased by a factor of 1.8 at CD and 1.5 at CQ during the same periods. Furthermore, similar diurnal variations were found for CO during polluted and clean periods. Thus, comparing the different enhancing factors between CO and other pollutants of interest can shed some light on the impact of non-meteorological factors on pollutants accumulation, as was done by using the CO-scaled pollutant concentrations.

13. Line 274-275: "CO can be considered as a reference pollutant species whose temporal variations were mainly from the impact of meteorological conditions." Why? See the comment 11. Also, I think the CO-scaling method should be further explained with more details and with references.

R: See our resposne to the previous comment. The CO-scaled pollutant concentration means the ratio of the pollutant concentration to CO concentration (e.g. PM2.5/CO, SO42-/CO, OC/CO, etc.). We have added the related reference (Zheng et al., 2015b, Zhang et al., 2015) in the revised manuscript.

14. Section 3.4.2: The diurnal trends of monitored gases could not give any solid evidence to support their hypothesis. In this case, especially when the formation mechanisms of secondary aerosols are discussed, high resolution data are necessary. We should not rely on the unsolid speculation.

R: We agree that high-resolution data would provide more and better information on the formation mechanisms of secondary aerosols. Unfortunately, such data cannot be obtained during the sampling campaign due to the lack of expensive on-line instruments. As explained n our response to general comment 3 above, even with daily data in-depth analyses can be conducted. In this study, Sulfur oxidation ratio (SOR) and nitrogen oxidation ratio (NOR) were defined to evaluate the degree of secondary transformation. Considering the low-resolution data, SOR and NOR were grouped according to temperature, RH and O3 concentration bins to explore the variation trends of SOR, NOR and SOC/OC. Such an analysis revealed that SO42- was predominantly formed through heterogeneous aqueous process while NO3- was formed by both homogeneous and heterogeneous reactions at both sites. The proposed formation mechanism of NO3- in the present study agreed with those found in an earlier study using high-resolution inorganic ions data (Tian et al., 2017).

Reference

Chen, Y., Xie, S.D., Luo, B., Zhai, C.Z., 2017. Particulate pollution in urban Chongqing of southwest China: Historical trends of variation, chemical characteristics and source apportionment. Science of the Total Environment 584, 523-534.

Gao, J.J., Tian, H.Z., Cheng, K., Lu, L., Zheng, M., Wang, S.X., Hao, J.M., Wang, K., Hua, S.B., Zhu, C.Y., Wang, Y., 2015. The variation of chemical characteristics of PM2.5 and PM10 and formation causes during two haze pollution events in urban Beijing, China. Atmospheric Environment 107, 1-8.

Hua, Y., Cheng, Z., Wang, S.X., Jiang, J.K., Chen, D.R., Cai, S.Y., Fu, X., Fu, Q.Y., Chen, C.H., Xu, B.Y., Yu, J.Q., 2015. Characteristics and source apportionment of PM2.5 during a fall heavy haze episode in the Yangtze River Delta of China. Atmospheric Environment 123, 380-391.

Li, H.Y., Zhang, Q., Zhang, Q., Chen, C.R., Wang, L.T., Wei, Z., Zhou, S., Parworth, C., Zheng, B., Canonaco, F., Prevot, A.S.H., Chen, P., Zhang, H.L., Wallington, T.J., He, K.B., 2017. Wintertime aerosol chemistry and haze evolution in an extremely polluted city of the North China Plain: significant contribution from coal and biomass combustion. Atmospheric Chemistry and Physics 17, 4751-4768.

Ming, L.L., Jin, L., Li, J., Fu, P.Q., Yang, W.Y., Liu, D., Zhang, G., Wang, Z.F., Li, X.D., 2017. PM2.5 in the Yangtze River Delta, China: Chemical compositions, seasonal variations, and regional pollution events. Environmental Pollution 223, 200-212.

Mozurkewich, M., 1993. The Dissociation-Constant of Ammonium-Nitrate and Its Dependence on Temperature, Relative-Humidity and Particle-Size. Atmospheric Environment Part a-General Topics 27, 261-270.

Paraskevopoulou, D., Liakakou, E., Gerasopoulos, E., Mihalopoulos, N., 2015. Sources of atmospheric aerosol from long-term measurements (5 years) of chemical composition in Athens, Greece. Science of the Total Environment 527, 165-178.

Pietrogrande, M.C., Bacco, D., Ferrari, S., Ricciardelli, I., Scotto, F., Trentini, A., Visentin, M., 2016. Characteristics and major sources of carbonaceous aerosols in PM2.5 in Emilia Romagna Region (Northern Italy) from four-year observations. Science of the Total Environment 553, 172-183.

Squizzato, S., Masiol, M., 2015. Application of meteorology-based methods to determine local and external contributions to particulate matter pollution: A case study in Venice (Italy). Atmospheric Environment 119, 69-81.

Tao, J., Gao, J., Zhang, L., Zhang, R., Che, H., Zhang, Z., Lin, Z., Jing, J., Cao, J., Hsu, S.C., 2014. PM2.5 pollution in a megacity of southwest China: source apportionment and implication. Atmospheric Chemistry and Physics 14, 8679-8699.

Tian, M., Wang, H. B., Chen, Y., Zhang, L. M., Shi, G. M., Liu, Y., Yu, J. Y., Zhai, C. Z.,

Wang, J., and Yang, F. M., 2017. Highly time-resolved characterization of water-soluble inorganic ions in PM2.5 in a humid and acidic mega city in Sichuan Basin, China, Sci Total Environ, 580, 224-234.

Wang, J., Li, X., Zhang, W.K., Jiang, N., Zhang, R.Q., Tang, X.Y., 2016. Secondary PM2.5 in Zhengzhou, China: Chemical Species Based on Three Years of Observations. Aerosol and Air Quality Research 16, 91-104.

Wang, P., Cao, J.J., Shen, Z.X., Han, Y.M., Lee, S.C., Huang, Y., Zhu, C.S., Wang, Q.Y., Xu, H.M., Huang, R.J., 2015. Spatial and seasonal variations of PM2.5 mass and species during 2010 in Xi'an, China. Science of the Total Environment 508, 477-487.

Wang, Q.Z., Zhuang, G.S., Huang, K., Liu, T.N., Deng, C.R., Xu, J., Lin, Y.F., Guo, Z.G., Chen, Y., Fu, Q.Y., Fu, J.S.S., Chen, J.K., 2015. Probing the severe haze pollution in three typical regions of China: Characteristics, sources and regional impacts. Atmospheric Environment 120, 76-88.

Zhang, Q., Quan, J. N., Tie, X. X., Li, X., Liu, Q., Gao, Y., and Zhao, D. L.: Effects of meteorology and secondary particle formation on visibility during heavy haze events in Beijing, China, Sci Total Environ, 502, 578-584, 10.1016/j.scitotenv.2014.09.079, 2015.

Zhao, P.S., Dong, F., He, D., Zhao, X.J., Zhang, X.L., Zhang, W.Z., Yao, Q., Liu, H.Y., 2013. Characteristics of concentrations and chemical compositions for PM2.5 in the region of Beijing, Tianjin, and Hebei, China. Atmospheric Chemistry and Physics 13, 4631-4644.

Zhao, X.J., Zhao, P.S., Xu, J., Meng, W., Pu, W.W., Dong, F., He, D., Shi, Q.F., 2013. Analysis of a winter regional haze event and its formation mechanism in the North China Plain. Atmospheric Chemistry and Physics 13, 5685-5696.

Zheng, B., Zhang, Q., Zhang, Y., He, K.B., Wang, K., Zheng, G.J., Duan, F.K., Ma, Y.L., Kimoto, T., 2015a. Heterogeneous chemistry: a mechanism missing in current models to explain secondary inorganic aerosol formation during the January 2013 haze

episode in North China. Atmospheric Chemistry and Physics 15, 2031-2049.

Zheng, G.J., Duan, F.K., Su, H., Ma, Y.L., Cheng, Y., Zheng, B., Zhang, Q., Huang, T., Kimoto, T., Chang, D., Poschl, U., Cheng, Y.F., He, K.B., 2015b. Exploring the severe winter haze in Beijing: the impact of synoptic weather, regional transport and heterogeneous reactions. Atmospheric Chemistry and Physics 15, 2969-2983.

---

## Author Response (AR1)

Dear Editor:

We have carefully revised the manuscript and addressed all of the comments provided by the two reviewers. The details can be found in our enclosed responses to the reviewers' comments. For your and the reviewers' convenience in reviewing the changes, a copy of the paper with track-changes is also attached below.

Thank you for taking care of the review process for this paper.

Sincerely,

Huanbo Wang and co-authors

**Response to Referee #2**

We greatly appreciate the helpful comments from the reviewer, which have helped us improve the paper. We have addressed all of the comments carefully, as detailed below. Our responses start with "R:".

**General Comments:**

This manuscript elucidates the seasonal variations, the formation mechanism, and the sources of  $PM_{2.5}$  in two megacities in Sichuan Basin. The concentrations of major chemical components of  $PM_{2.5}$  in different seasons are investigated. The chemical characteristics of clean days and polluted days are presented to study the formation of key chemical species as well as transportation pathway of secondary aerosols. In general, the research results provide meaningful information on both formation mechanism and environmental control strategies of  $PM_{2.5}$ . However, there are still some key issues which need to be addressed before possible publication.

R: We have carefully studied the several key issues raised by this and another reviewer and revised the paper accordingly. We hope the reviewer will find that the quality of the paper has been improved significantly.

1. The factors contributing to the different temporal patterns of sulfate and nitrate should be further discussed. Specific heterogeneous reaction which may play important roles in polluted days and its major contributing components can be clearly pointed out.

**R**: We have added information of the gaseous precursors and meteorological parameters to facilitate the more in-depth discussion on the temporal patterns of  $SO_4^{2^-}$  and  $NO_3^-$  in the revised manuscript. We agree with the reviewer that heterogeneous reactions likely contributed to the formation of secondary aerosols considering the high relative humidity (RH) (60-88%) conditions during the study campaign. In the revised paper, the formation of secondary aerosols through heterogeneous reactions was discussed in detail based on the relationships between SOR, NOR and SOC/OC and RH. Note that RH exhibited no significant difference between clean and polluted periods in our study, suggesting RH was not the driving force for the polluted episodes in Sichuan Basin. This phenomenon was different from what was found in eastern coastal China and North China Plain, where sharp RH increase was observed during polluted episodes.

2. The discussion on  $PM_{2.5}$  formation process can be combined with the analysis on the variation of gaseous precursors, including  $SO_2$  and  $NO_2$ .

**R**: As mentioned above, information of gaseous precursors has been added in the revised paper. Temporal variations of  $PM_{2.5}$  and major gaseous precursors were then

discussed together. For example, gaseous precursors (SO2 and NO2) and secondary inorganic aerosols (SO42- and NO3-) increased by a factor of 1.5-3.7 during the polluted periods than clean periods. To exclude the influence of atmospheric physical processes on these variations, CO-scaled variations were also provided. The CO-scaled SO42- was 40-70% higher and that of NO3- was 80-120% higher during the polluted periods, while their respective gaseous precursors were no more than 30% higher. These numbers suggested stronger chemical transformation from gaseous precursors to PM2.5 during the polluted periods, as well as the significant contribution of meteorological condition to the high PM2.5 levels.

3. More discussion on geographical sources of different chemical components of  $PM_{2.5}$  is recommended. I also suggest the analysis on the different topography of these two megacities, which will help to better explain the impact of local emission and regional transportation.

**R**: We have plotted several PSCF maps to illustrate the geographical sources of the different chemical components in  $PM_{2.5}$  and their gaseous precursors. Generally, similar spatial distributions of potential sources for  $PM_{2.5}$  and its chemical components were observed. At the CD site, high concentrations of the pollutants were mostly associated with sources broadly located in the southeast of the basin, covering Neijiang, Zigong, Yibin, Luzhou and east part of Chongqing. At the CQ site, the northeast area of Chongqing was identified as the major sources.

The impact of the special topography and meteorological conditions on  $PM_{2.5}$  levels at the two sampling sites were also discussed in detail. Firstly, CD and CQ sampling sites are both located in the Sichuan Basin, which is surrounded by high mountains. Such topography often forms a barrier for the dispersion of pollutants and causes air stagnation within the basin, therefor facilitates regional-scale pollution episodes inside the Basin. Back trajectory analysis showed that air masses reaching CD and CQ only traveled for short distances, primarily within the Sichuan Basin. This highlighted the impact of the special topography on  $PM_{2.5}$  pollution at the two sites. Secondly, Chongqing is a famous mountain city with the majority of population using motorcycle instead of bicycle as the main daily travel tools. This contributed more VOCs emissions and high OC concentrations at CQ than CD.

Besides, I have some specific comments on the manuscript as follows: 1. Section 2.5: I'd like to recommend adding detailed equations of PSCF analysis for better understanding.

**R**: Information added.**

2. Line 170-171: Citation format error: "Tao et al. (2013, 2014)" should be corrected into "(Tao et al., 2013, 2014)".

**R**: Corrected.**

3. Line 244: The authors claimed the concentration of  $NO_3^-$  decreased on the polluted days in the warm season of CQ. But in Figure 6(d), the concentration of  $NO_3^-$  is higher in the polluted days. There seems to be contradictory.

**R**: Thanks for pointing out this negligence, which has been corrected in the revised paper.

4. Section 3.4.2: The authors applied CO-scaled  $PM_{2.5}$  and major components to isolate the impact of meteorological conditions. Specific scaling approach or related references should be provided.

**R**: The CO-scaled pollutant concentration was calcualetd as the ratio of concentrations between a pollutant of interest and CO (e.g.  $PM_{2.5}/CO$ ,  $SO_4^{2-}/CO$ , OC/CO). We have added explanaiton and related references (Zheng et al., 2015, Zhang et al., 2014) in the revised paper.

5. Section 3.4.3: I'd like to recommend adding a graph containing RH levels and NO3- concentration between the two sites. Also, a correlation between  $[NO_3^-]/[SO_4^{2-}]$  and  $[NH_4^+]/[SO_4^{2-}]$  is suggested to investigate the difference of NO3- formation between CQ and CD.

**R**: NO3- concentrations at CD and CQ were compared for all the seasons. NO3- was 58% lower at CQ than CD in summer, but was at similar levels in the other seasons. NO2 and most meteorological parameters (except RH) were comparable at both sites in summer. Thus, the lower NO3- at CQ in summer was likely mainly caused by the lower RH, which inhibited the formation of NH4NO3. We have added a figure in the supplemental information document showing the temporal variations of ambient RH and deliquescence relative humidity (DRH), which was used to explain the different NO3- concentrations between CD and CQ in summer.

We have also added correlation analysis between  $[NO_3^-]/[SO_4^{2^-}]$  and  $[NH_4^+]/[SO_4^{2^-}]$  and related in-depth discussion on the formation mechanism of  $NO_3^-$  in winter in the revised paper.

6. Figure 3: The black dots which indicate the average values should be stated in the figure caption.

**R**: We have added the description of the legend in this figure.

Reference cited above:

Zheng, G. J., Duan, F. K., Su, H., Ma, Y. L., Cheng, Y., Zheng, B., Zhang, Q., Huang, T., Kimoto,

T., Chang, D., Poschl, U., Cheng, Y. F., and He, K. B.: Exploring the severe winter haze in Beijing: the impact of synoptic weather, regional transport and heterogeneous reactions, Atmos Chem Phys, 15, 2969-2983, 10.5194/acp-15-2969-2015, 2015.

Zhang, Q., Quan, J. N., Tie, X. X., Li, X., Liu, Q., Gao, Y., and Zhao, D. L.: Effects of meteorology and secondary particle formation on visibility during heavy haze events in Beijing, China, Sci Total Environ, 502, 578-584, 10.1016/j.scitotenv.2014.09.079, 2015.

**Response to Referee #3**

We greatly appreciate the helpful comments from the reviewer, which have helped us improve the paper. We have addressed all of the comments carefully, as detailed below. Our responses start with "R:".

The manuscript is based on the observation conducted in four selected months in two cities in the Sichuan Basin, China. It represents the results of  $PM_{2.5}$  and the chemical components. The seasonal variations are shown and the difference in terms of the formation mechanisms and geographical influence between the two cities is discussed.

The content of this manuscript fits the scope of ACP and the data is interesting to be studied. However, I found this manuscript is only a report of the results in a rarely investigated region in China but without in-depth analysis. No novel point has been raised and discussed in this manuscript. I would not recommend it to be published in ACP in the current stage.

**R:** Based on this and another reviewer's critical comments, we have added many in-depth analyses in the revised manuscript. These include: (1) providing gaseous precursors and meteorological parameters data to explain the seasonal variation trends of  $PM_{2.5}$  and its chemical components; (2) conducting air mass back trajectory analyses to illustrate the influence of topography on  $PM_{2.5}$  pollution in Sichuan Basin; (3) adding deliquescence relative humidity (DRH) analysis for the summer season to explain the different NO3- concentrations at the two sites, and (4) comparing the characteristics of  $PM_{2.5}$  pollution episodes in Sichuan Basin with those in the other regions of China.

The quality of the paper has been substantially improved, as demonstrated by the many novel findings. A few major ones are detailed below.

(1) The study identified different driving mechanisms for the polluted PM2.5 episodes in the Sichuan Basin than in the other regions of China. For example, sharply increased relative humidity (RH) was thought to be one of the main factors causing high inorganic aerosol concentrations during the polluted periods in eastern coastal China and North China Plain where mostly with flat terrain (Li et al., 2017; Zhao et al., 2013; Zheng et al., 2015a; Zheng et al., 2015b, Gao et al., 2015; Hua et al., 2015; Wang et al., 2015). In contrast, RH did not differ much between polluted and clear periods in Sichuan Basin. Instead, the special topography and meteorological conditions in this region resulted in the polluted PM2.5 levels, and different local topography between CD and CQ further added different pollution characteristics between the two sites. Note that Sichuan Basin is completely surrounded by high mountains and constantly characterized by low wind speeds. RH in the region is high throughout the year, which is conducive for heterogeneous reactions forming

**inorganic aerosols.**

(2) The study identified the sub-regional characteristics of emission sources. An additional VOCs emission source was identified for CQ than CD based on higher OC concentrations at CQ. This additional source was attributed to motorcycle traffic in CQ since it is a famous mountain city where most people use motorcycle as daily traffic tools. According to the Chongqing Statistical Yearbook 2015, the number of motorcycles was 2.0 million (among the total of 2.3 million motor vehicles) in 2014, which was much higher than those (0.7 million) in Chengdu (Chengdu Statistical Yearbook 2015).

(3) The study identified sub-regional characteristics of inorganic aerosols. Although the whole Sichuan Basin (as the regional scale) was characterized by special topography (surrounded by mountains) and hot and humid air, there were sub-regional differences within the basin, as contrasted between the two largest cities (CD and CQ) in this region. Thus, different aerosol characteristics were found at the two sites. For example, the lower NO3- at CQ than at CD was identified to be caused by the lower RH based on the deliquescence relative humidity (DRH) of NH4NO3 (Mozurkewich, 1993).

With these additoanl in-depth analyses and inovative results, we hope the reviewer will find ithe paper meets the standard of the jorunal.

General comments:

1. The sampling campaign in four selected months may not be enough to provide sufficient data to answer the questions (objectives) which are supposed to be studied in this work. The two sampling sites seems not ideal to understand the characteristics of  $PM_{2.5}$  in two basined cities with typical geographical features. Regional sites without direct emissions are better in my opinion. In order to discover and reveal the formation mechanisms of secondary aerosols, more data and analysis are necessary.

**R**: Based on existing literature, the four month data observed in four typical seasons should be enough for exploring the seasonal and annual patterns of aerosol pollution levels and for exploring potential formation mechanisms, as has been demonstrated in studies for other regions of China as well as for other countries (Paraskevopoulou et al., 2015; Pietrogrande et al., 2016; Squizzato and Masiol, 2015; Ming et al., 2017; Tao et al., 2014; Wang et al., 2016; Wang et al., 2015; Zhao et al., 2013).

We agree with the reviewer that a traditional regional background site would choose a rural site far away from urban areas/local emissions in order to study the background pollution levels. However, this is not the focus of our study. Emission control policies aim to reduce PM2.5 pollution for populated regions for human health concerns. In this sense, urban areas, especially in megacities, are the major concerns. Chengdu and

Chongqing are the two biggest cities in Sichuan Basin. Thus, emission sources, transportation and chemical transformation of atmospheric aerosols observed in these cities should be investigated thoroughly. The two monitoring sites selected in this study should represent the typical urban environment in their respective cities (Chen et al., 2017; Tao et al., 2014). Although none of the two sites alone would represent the whole region of the Sichuan basin, the similarities between the two sites should represent the reginal-scale characteristics of urban-environment pollution while the differences between the two sites should reflect the sub-regional characteristics of urban pollution. This reviewer seemed to have a different view of the regional-scale phenomenon than what we had in mind, which we understand, was due to the different considerations. The selection of our sites perfectly served the goals of our study.

Although the current data set is not very big, it is enough to provide many in-depth analyses, as we have done in the revised paper after incorporating both reviewers' recommendations (with more details in our responses to the specific comments below).

2. The data are not well presented in this manuscript. The readers can hardly find the sufficient information to know and understand the results. For example, how many samplers were collected in the campaign? How many samples were taken and how about the variations of data in clear days, moderate polluted days and heavy polluted days? Were there some different pollution episodes?

**R**: We have added more details about the sampling information in the revised manuscript. A total of 112 samples were collected on daily basis at each site, which were 27, 28, 28 and 29 days in autumn, winter, spring and summer, respectively. To view the pollution episodes clearly, we have added a figure describing the temporal variations of daily  $PM_{2.5}$  concentrations, gaseous precursors and meteorological parameters during the entire study periods. The number of polluted days was 8, 21, 4 and 1 in autumn, winter, spring and summer (total 34 days), respectively, at CD, and was 4, 19, 6 and 2 (total 31 days) at CQ.

3. The analysis and discussion are superficial and full of speculation. No solid evidence can be provided to support the conclusions, which makes the significance and implication ungrounded. For example, to support their hypothesis, the diurnal variations of monitored gases are presented and discussed. However, the data of  $PM_{2.5}$  and their chemical components are on daily basis, which weaken the analysis and leads to vague conclusions.

**R**: We agree that it is ideal to have hourly aerosol data for more detailed analyses, which unfortunately could not be obtained in this study due to instruments limitations. However, in-depth analyses can be done using daily data as demonstrated in earlier studies as well as in our revised version of this manuscript (more details below in our

**responses to specific comments).**

More specific comments are shown as follows:

1. As I suggested above, are the two sampling sites and the data representative for this investigation on the characteristics of aerosol in the two basined cities? Obviously they are both highly affected by the traffic emissions which may bias the analysis. The topography of the two sites and the influence should be discussed.

**R**: See our responses to point 1 of general comments above regarding regional representative of the sites. We have added discussion on topography related impacts in the revised paper. Briefly, CD is located in the west while CQ on the eastern margin of Sichuan Basin. The basin is surrounded by mountains in all directions, which forms a barrier for pollutants dispersion and thus causes frequent stagnant air in the basin. This results in regional-scale pollution episodes in Sichuan Basin. Air mass back trajectory analysis showed that air masses reaching CD and CQ only traveled for short distances and primarily within the basin. Such a phenomenon highlighted the impacts of the special topography on  $PM_{2.5}$  pollution.

2. Line 78: Please provide the details of the sampler. Three samplers were used in this campaign. The comparison of the three samplers should be provided to show the accuracy and consistency of the data.

**R**: The samplers used in this study are described in the manuscript. At CD site,  $PM_{2.5}$  sampling was carried out using a versatile air pollutant sampler (URG Corp., URG-3000K, North Carolina, USA), which has three channels. One channel was used to load  $PM_{2.5}$  sample on Teflon filter for mass and trace elements anlysis and the other one was equipped with quatz filter for water-soluble inorganic ions and carbonaceous components analysis. At the CQ site, a low-volume aerosol sampler (BGI Corp., frmOmni, USA) operating at a flow rate of 5 L min-1 was used to collect  $PM_{2.5}$  samples on Teflon filter, and another sampler (Thermo Scientific Corp. Partisol 2000i, USA) with a flow rate of 16.7 L min-1 was used to collect  $PM_{2.5}$  samples on two parallel channel at CD site and the simultaneous sampling on two instruments at CQ site allowed the contemporary chemical determination of the loading  $PM_{2.5}$ .

The three samplers used in this study were commercial instruments and widely used in  $PM_{2.5}$  sampling. We examined the flow rate of each sampler before and after sampling carfully to ensure the quality of sampling.

3. How many samples were collected? How the blank filters (lab blank and field blank) were collected?

R: See our responses to point 2 of general comments. Field blanks were collected

every two weeks in each season, resulting 8 filed blanks at each site. In order to check the background contamination from the laboratory, three lab blank filters in each campaign were stored in a clean Petri slides in the dark and were analyzed the same ways as the sampling filters. The detailed information has been added in the revised manuscript.

4. Line 111-113: There were only 5 elements detected by XRF. Normally I would expect more elements could be measured by the XRF technique. Why?

**R**: Besides the 5 crustal elements (Al, Si, Ca, Fe and Ti) used in the study, another 16 elements including Na, Mg, S, Cl, K, V, Cr, Mn, Co, Ni, Cu, Zn, Rb, Sb, Ba and Pb were also determined by the XRF method. Among those elements, Na, Mg, Cl, S and K were discussed in the form of ions, and other metal elements accounted for less than 1% of the  $PM_{2.5}$ . Thus, the above 16 elements were not considered in identifying the major chemical components that were responsible for the  $PM_{2.5}$  pollution.

5. Line 121: Please provide the details of the weather station.

**R**: We have added some details of the weather station in the revised manuscript.

6. Line 178-179: The authors pointed out that higher sulfate concentrations were found in summer. In Table 1, I found that lower sulfate average was in summer than that in winter. Please check the data.

**R**: We have clarified the explanation.  $SO_4^{2-}$  was the highest in winter, but not the lowest in summer. In contrast, many other pollutants had the lowest in summer.

7. Line 178-185: The discussion on sulfate, nitrate, chloride and potassium seems superficial and arbitrary. The analysis should be based on the data from this campaign and be made with in-depth study instead of guesses.

**R**: We have rewritten this section completely with additional data of meteorological parameters and gaseous precursors. See our summary responses at the top of this file.

8. Line 188-190: The high SOC content was observed in winter. In this work, the estimation of SOC mainly depends on the seasonal minimum of OC/EC. However, it should not be surprise to see high OC in winter because organic aerosols may not necessarily be only formed by secondary reaction but also by direct emissions (e.g. biomass burning).

**R**: We are aware of the limitations of approach, but think it is a practical method for estimating SOC, as has frequently been used in literature. We noted that there was no extensive coal combustion or wood burning for domestic heating in winter due to the warm climate in this region. Therefore, biomass burning should have small effects on

**OC concentrations.**

9. Section. 3.3 discusses the difference of data between the two sites. As it known to all, the difference can be due to many possible factors (emissions, atmospheric reactivity, meteorological conditions, the surrounding terrains). It is really hard to synthesize significant information from the comparison. Therefore, more in-depth studies are necessary.

**R**: We have identified major factors causing the differences between the two sites through the following in-depth analyses, such as back trajectory analyses and deliquescence relative humidity analyses (see some details in our summary responses at the top of this file).

10. Line 227-238: More information should be provided for the pollution episodes. For example, how many polluted days and in which seasons were captured? How many pollution episodes were observed?

**R**: We have added a figure and related information in the revised paper (also see response to point 2 of general comments above).

11. Line 254-256: The distinct characteristics in the urban area in the Sichuan Basin should be further investigated and discussed. How may the topography and meteorological conditions influence on the characteristics?

**R**: See our summary responses at the top of this file.

12. Line 271-272: "Both CO and EC concentrations increased on polluted days, suggesting the important role the meteorological condition played on  $PM_{2.5}$  accumulation." Why? I cannot see any link. The occurrence of CO and EC in the troposphere should be influenced by the emissions, removal mechanisms and other factors (including meteorological conditions but not exclusively).

**R**: Such a conclusion was based on this hypothesis: the variations of CO were mainly controlled by meteorological factors (Zheng et al., 2015b; Zhang et al., 2015) while those of the other pollutants were by both meteorology and chemical transformation. In this study, PM2.5 and gaseous precursors increased by a factor of 1.8-3.3 during polluted periods than clean periods while CO only increased by a factor of 1.8 at CD and 1.5 at CQ during the same periods. Furthermore, similar diurnal variations were found for CO during polluted and clean periods. Thus, comparing the different enhancing factors between CO and other pollutants of interest can shed some light on the impact of non-meteorological factors on pollutants accumulation, as was done by using the CO-scaled pollutant concentrations.

13. Line 274-275: "CO can be considered as a reference pollutant species whose

temporal variations were mainly from the impact of meteorological conditions." Why? See the comment 11. Also, I think the CO-scaling method should be further explained with more details and with references.

**R**: See our response to the previous comment. The CO-scaled pollutant concentration means the ratio of the pollutant concentration to CO concentration (e.g.  $PM_{2.5}/CO$ ,  $SO_4^{2-}/CO$ , OC/CO, etc.). We have added the related reference (Zheng et al., 2015b, Zhang et al., 2015) in the revised manuscript.

14. Section 3.4.2: The diurnal trends of monitored gases could not give any solid evidence to support their hypothesis. In this case, especially when the formation mechanisms of secondary aerosols are discussed, high resolution data are necessary. We should not rely on the unsolid speculation.

**R**: We agree that high-resolution data would provide more and better information on the formation mechanisms of secondary aerosols. Unfortunately, such data cannot be obtained during the sampling campaign due to the lack of expensive on-line instruments. As explained n our response to general comment 3 above, even with daily data in-depth analyses can be conducted. In this study, Sulfur oxidation ratio (SOR) and nitrogen oxidation ratio (NOR) were defined to evaluate the degree of secondary transformation. Considering the low-resolution data, SOR and NOR were grouped according to temperature, RH and O3 concentration bins to explore the variation trends of SOR, NOR and SOC/OC. Such an analysis revealed that  $SO_4^{2-}$  was predominantly formed through heterogeneous aqueous process while NO3- was formed by both homogeneous and heterogeneous reactions at both sites. The proposed formation mechanism of  $NO_3^-$  in the present study agreed with those found in an earlier study using high-resolution inorganic ions data (Tian et al., 2017).

*Correspondence to:* Fumo Yang (fmyang@cigit.ac.cn)

| 1  | Abstract. To investigate the characteristics of $PM_{2.5}$ and its major chemical components, formation                                          |
|----|--------------------------------------------------------------------------------------------------------------------------------------------------|
| 2  | mechanisms, and geographical origins in the two biggest cities, Chengdu (CD) and Chongqing (CQ), in                                              |
| 3  | Sichuan Basin, daily PM 2.5 samples were collected simultaneously at one urban site in each city for four                             |
| 4  | consecutive seasons from October autumn 2014 to July summer 2015. Annual mean concentrations of                                                  |
| 5  | $PM_{2.5}$ were $67.0 \pm 43.4$ and $70.9 \pm 41.4 \ \mu g \ m^{-3}$ at CD and CQ, respectively. Secondary inorganic                             |
| 6  | aerosols (SNA) and organic matter (OM) accounted for 41.1% and 26.1%, respectively, of PM 2.5 mass                                    |
| 7  | at CD, and 37.4% and 29.6% at CQ, respectively. Seasonal variations of PM 2.5 and its-major chemical                                  |
| 8  | components were significant, usually with the highest mass concentration values in winter and the                                                |
| 9  | lowest in summer. Daily PM 2.5 concentration exceeded the national air quality standard on 30% of the                                 |
| 10 | sampling days at both sites, and most of the pollution events were at the regional scale within the basin                                        |
| 11 | formed by stagnant meteorological conditions. The concentrations of carbonaceous components were                                                 |
| 12 | higher at CQ than CD, likely partially caused by emissions from the large amount of motorcycles in CQ.                                           |
| 13 | SNA and OM concentrations on polluted days were 12.7-3.4 3 times higher those on polluted days than-                                             |
| 14 | on clean days at both sites, whereas their percentage contributions to PM 2.5 -mass varied differently-                               |
| 15 | among the components and between the two sitescities. Homogeneous Gasgas-phase oxidation-                                                        |
| 16 | reactions probably played an important role on-in the formation of SO42- , while both homogeneous and |
| 17 | heterogeneous reactions contributed to the formation of NO3 - .secondary aerosols when PM2.5-mass-                                    |
| 18 | varied in the range of 75-150 µg m -3 , while heterogeneous transformation was likely the major-                                      |
| 19 | mechanism on the heavy polluted days. Geographical origins of emissions sourcesregions causing-                                                  |
| 20 | contributing to high PM2.5 masses at both sites were identified to be mainly distributed within the basin                      |
| 21 | at both sites based on potential source contribution function (PSCF) analysis.                                                                   |
| 22 |                                                                                                                                                  |

**24 **1 Introduction**

Fine particles ( $PM_{2.5}$ , particulate matter with an aerodynamic diameter smaller than 2.5  $\mu$ m) have

- adverse effects on human health (Anderson et al., 2012;Lepeule et al., 2012;Taus et al., 2008),
- deteriorate air quality (Zhang et al., 2008;Paraskevopoulou et al., 2015), reduce atmospheric visibility
- 28 (Fu et al., 2016;Cao et al., 2012;Baumer et al., 2008), impact climate (Ramanathan and Feng,
- 29 2009;Hitzenberger et al., 1999;Mahowald, 2011), and affect ecosystem (Larssen et al., 2006). In the past
- two decades, China has experienced serious  $PM_{2.5}$  pollution due to the rapidily increasing energy
- 31 consumption through econmic development, industrialization and urbanization (Tie and Cao, 2009; Tao

32 et al., 2017). The National Ambient Air Quality Standards (NAAQS) for PM2.5 was promulgated by the

- Chinese government in 2012, and strict strategies have been implemented nationwide, e.g. controling
- 34 SO2 emissions by installing desulphurization system in coal-fired power plants and conversion of fuel to
- natural gas (Lu et al., 2011), mitigating  $NO_x$  emissions through traffic restrictions, and reducing
- biomass burning through straw shredding. Despite these efforts, there are still many cities that have not yet\_met the current NAAQS (Tao et al., 2017). According to the '2013-2015 Reports on the State of Environment of China', annual mean concentration of  $PM_{2.5}$  in all the\_74 major cities over China was 72, 64, and 50 µg m-3 in 2013, 2014 and 2015, respectively, and only 4.1%, 12.2% and 22.5% of the monitored cities met the NAAQS (35 µg m-3).

41 Previous studies showed that Beijing-Tianjin-Hebei area (BTH), Yangtze River Delta (YRD), Pearl River Delta (PRD), and Sichuan Basin were the four main regions in China with severe aerosol 42 pollution (Tao et al., 2017). While many studies have been conducted in BTH, PRD and YRD regions 43 to understand the general characteristics of PM2.5 and its chemical components, formation mechanism, 44 45 and sources (Ji et al., 2016;Li et al., 2015;Quan et al., 2015;Tan et al., 2016;Yang et al., 2015;Zhang et al., 2013;Zhao et al., 2015;Zhao et al., 2013a;Cheng et al., 2015;Zheng et al., 2015a;Yang et al., 2011a), 46 only a few studies have focused on Sichuan Basin (Tao et al., 2014; Tian et al., 2013; Yang et al., 2011b). 47 Covering an area of 260,000 km2 and with a population of around 100 million in southwest China, the 48 Sichuan Basin is a subtropical expanse of low hills and plains and the most populated basin in China. It 49 is a subtropical expanse of low hills and plains and is completely encircled by high mountains and 50 plateaus, and It is also characterized by persistently high relative humidity, and extremely low wind 51 speeds all the year-round (Guo et al., 2016; Chen and Xie, 2013). It is supposed that tThe characteristics 52 of PM2.5 in Sichuan Basin are supposed toshould be very different from those in eastern coastal China 53

(i.e. PRD and YRD) and North China Plain (i.e. BTH) due to its-the special topography and 54 meteorological conditions, besides emission sources, in the basin. More specificallyFurthermore, the 55 terrain inof the two megacities in the basin are is also distinct from each other significantly, i.e., 56 Chongqing is a mountainous citymunicipality lying on the eastern margin of the basin while Chengdu is 57 located in a completely a flat city on the western margin of the basin. Therefore, there is a great interest 58 59 in comparing the chemical components of PM2.5 and characterizing pollution episodes between in 60 Chengdu and Chongqingthe two cities. The present study aims to fill this gap by measuring chemically-resolved PM2.5 in Chengdu and 61 Chongqing in four consecutive seasons during 2014-2015. The main objectives are to: (1) characterize 62 the seasonal and site differences of PM2.5 mass and its major chemical components at between the two-63 urban sites in Sichuan Basinin urban environemntss of Chengdu and urban-Chongqing; (2) compare the 64 PM2.5 chemical components compositions under different PM2.5 pollution levels and identify the major 65 chemical components that are responsible for long-lasting PM2.5 pollution episodes in winter; (3) 66 explore the possible formation mechanism of the secondary organic and inorganic aerosols; and (4) 67 reveal the geographical source regions causing contibuting to the high PM2.5 levels through potential 68 source contribution function (PSCF) analysis. The comparison of the differences of PM2.5 in this paper-69

provides insights regarding the extent to which fine particulatePM2.5 pollution can vary in terms of inchemical composition, formation mechanisims, and geographical origins between the two megacities ina great basin. This information has implications for better understanding the reasonsKnowledge gained

in this study provides scientific basis for making future emision control plocies aiming to allivating –

75 **2 Methodology**

for heavy haze PM2.5 pollution in this unique basin.

70

71

72

73

74

**76 2.1 Sampling sites**

[revised manuscript text omitted]

**177 **3 Results and discussion**

**178 **3.1** Overview of PM2.5 mass concentrations and major components chemical composition**

**179 **3.1.1 Overview**

Table 1 presents seasonal and annual mean concentrations of PM2.5 and its major chemical components 180 at CD and CQ during the sampling periods. Daily  $PM_{2.5}$  ranged from 11.6 to 224.7 µg m-3 with annual 181 average being 67.0  $\pm$  43.4 µg m-3 at CD site and 70.9  $\pm$  41.4 µg m-3 at CO site. The annual average 182 values, which were about two times of the NAAQS annual limit. Secondary inorganic aerosol (SNA, 183 the sum of  $SO_4^{2-}$ ,  $NO_3^{-}$  and  $NH_4^{+}$ ) and carbonaceous species together represented more than 70% of 184  $PM_{2.5}$  mass at both sites (Fig. 2). The annual mean concentrations of the total SNA were 27.6  $\mu$ g m-3 at 185 CD and 26.5  $\mu$ g m-3 at <del>CD and CQ, respectively</del>, contributing 41.1% and 37.4% of to PM2.5 mass, 186 respectively. Among these At CD, SO42-, NO3- and NH4+ were 11.2, 9.1, and 7.2 µg m-3, respectively, 187 oraccounteding for 16.8%, 13.6% and 10.8%, respectively, of PM2.5 mass at CD, and 12.2, 7.7 and 6.6 188 µg m-3 orwhile at CQ the corresponding percentages were-17.2%, 10.9% and 9.2%, respectively, of-189 PM2.5-mass at CQ. Organic matters (OM), estimated from OC by-using a conversion factor of 1.6 to 190 account for other elements presented in organic compounds (Turpin and Lim, 2001; Tao et al., 2017), 191 were the most abundant species in PM2.5, accounting for 26.1% and 29.6% of PM2.5 mass at CD and 192 CQ, respectively. In contrast, EC only comprised of around 6% at both sites. The annual mean 193 concentrations of OC and EC at CQ were 20% and 25%, respectively, higher at CQ than those at CD. 194

195 The annual mean concentration of Fine fine soil (FS), can be estimated calculated by summing the oxides of the major crustal elements mainly associated with soil, i.e., Al2O3, SiO2, CaO, FeO, Fe2O3, 196 and TiO2 (Huang et al., 2014), The annual mean concentration of FS at CQ was 6.7 µg m-3 (or 9.5% of 197  $PM_{2.5}$  mass) at CQ, which. It is noted that this was about two times of that at CD (3.8 µg m-3 or, 5.7% of 198  $PM_{2.5}$  mass). The minor components such as K+ and Cl- constituted less than 5% of  $PM_{2.5}$ . The 199 200 unaccounted portions of PM2.5 reached 18.3% at CD and 15.3% at CQ, which wereas likely related to 201 the uncertainties in the multiplication factor used for estimatingon of OM and FS, other unidentified 202 species, and measurement uncertainties.

**203 **3.1.2 Seasonal variations**

Figure 3 shows the seasonal variations of PM2.5 and its major chemical 204 components at CD and CQ-sites. Seasonal variations of any pollutants should be influenced by the 205 seasonal variations of n source emission intensities, atmospheric processes and meteorological 206 conditionsparameters. Unlike in northern China, Tthere wereas no significant extensive coal combustion 207 or wood burning utilization for domestic heating in winter due to the warmhigher temperature (around 208 209 10°C on average) in Sichuan Basin, hence atmospheric processes and meteorological conditions played 210 a vital role in the seasonal variations of PM2.5. On a Seasonal basis, average PM2.5 mass was the 211 highest in winter at both sites CD and CQ, which was 1.8-2.5 times of those in the other seasons. In 212 contrast, Sits seasonal differences among the other three seasons were generally small, i.e.g., less than 40%. Besides the high emissions of  $SO_2$  and  $NO_*$  in winter, stagnant Stagnant air condition with 213 frequent calm winds and low boundary planetary boundary layer height layer height was the another 214 major cause of the highest PM2.5 mass in this seasonwinter (Table 1) (Liao et al., 2017; Chen and Xie, 215 216 2013). Seasonal differences among the other three seasons were generally small, e.g., less than 40%. All the major PM2.5 components but except FS followed the seasonal pattern of PM2.5 mass with-217 the highest concentrations in winter, but with subtle differences. The highest FS concentrations were 218 219 observed in spring at both sites. The relatively higher wind speeds and lower RH in spring were 220 conducive for re-suspension of crustal dust and resulted in higher FS concentrations. In addition, thefrequent spring dust storms originated in the northwestern China was able to reach Sichuan Basin via 221 long-rang transport, and would cause the elevated FS concentrations (Chen et al., 2015;Tao et al., 2013). 222 The majority of the PM2.5 components showed a summer minimum, but not  $SO_4^2$ . The minimum-223 224 of OC at CD and FS at CQ appeared in spring and autumn, respectively. The which was caused by high

225 temperature and planetary boundary layer height favoringed the pollutants dispersion in summer. In-

226 addition, and abundant precipitation favoring wet scavengingin summer could scavenge the air

227 pollutants through wet deposition and in turn decreased the pollutant levels in the air. HigherIn contrast,

228 the lowest seasonal average concentrations did not appear in summer for  $SO_4^{2-}$  concentrations insummer were-likely due to the enhanced photochemical reactions associated with higher temperature
 and stronger solar radiation in summer. High O3 concentrations in summer also supported this seasonal
 trend.

It is also noted that the seasonal variations inof  $NO_3^-$  were much larger than those in of  $SO_4^{2^-}$  and NH4+, which. This can be explained by the enhanced formation of NO3- under high RHrelativehumidity in winter, and volatility of NH4NO3 in summer under high temperature condition (Pathak et al., 2009;Quan et al., 2015;Squizzato et al., 2013). In addition, thermodynamically driven behavior of NH4NO3 was another factor for the lower NO3- concentrations in summer (Wang et al., 2016;Kuprov et al., 2014). High levels of CI- and K+ in winter and of FS in spring should be caused by biomass burningor spring dust storms (Tao et al. 2013, 2014).

239 Both OC and EC showed exhibited the highest concentrations in winter at CD and CQ, whereas seasonal differences of those carbonaceous components were less distinct in other seasons, e.g. the 240 241 variations of OC and EC among the other three seasons were less than 30%. Seasonal average SOC was unexpectedly the highest in winter at both sites, different from the anticipated high value in summer in 242 243 consideration of strong photochemical reaction. Condensation of semi-volatile organic aerosols in winter seemed to play a larger role than photochemical reaction in summer, knowing that low 244 temperature favors condensation process (Sahu et al., 2011;Cesari et al., 2016). Although high O3 and 245 strong solar radiation condition in summer was conducive to strong photochemical reactions, high 246 247 temperature favoreds gas-particle partitioning in the gaseous phase and thus limited the increase of SOC (Strader et al., 1999). 248

The sS easonal average relative contributions of major chemical components to  $PM_{2.5}$  are showndepicted in Fig. 2. The seasonal average contributions of SNA to  $PM_{2.5}$  only varied within a small range from 39.5% to 43.2% at CD, whereas in a relatively larger range from 31.0% in summer to 37.1-41.5% in other seasons at CQ. The smaller contribution in summer at CQ was mainly due to the lower  $NO_3^$ concentrations. At both CD and CQ,  $NO_3^-$  and  $NH_4^+$  showed the highest contributions in winter and the lowest ones in summer, whereas an opposite trend was found for  $SO_4^{2-}$ . The contributions of carbonaceous components (the sum of OC and EC) generally followed the seasonal patterns of SNA, accounting for 26.7-38.8% of  $PM_{2.5}$  mass. Among these, OM showed the lowest fractions in  $PM_{2.5}$  in spring (21.1%) at CD and highest value in winter (33.6%) at CQ, while the percentages of OM in other seasons were similar at both sites, around 27%. The seasonal variations of EC fractions were not obvious, with a slightly higher value in spring. The highest contributions from FS was more than 10%, appeared in spring at both sites.

**261 3.1.3 Similarities and differences between the two sites**

Although none of the two sites alone can represent the whole region of the Sichuan basin, the similarities in the characteristics of the major pollutants between the two sites should represent the reginal-scale characteristics of urban-environment pollution while the differences between the two sites should reflect the sub-regional characteristics of urban pollution. A comparison between the two sites in terms of seasonal-average concentrations of major chemical components isare shown in Fig. 4 and discussed in detail below.

Despite the 260 km distance between the two sampling sites, a moderate similarity was observed in 268 autumn, winter and spring on the basis of low COD values (0.15-0.18), indicating the regional-scale 269 270 PM2.5 pollution pattern in Sichuan Basin and the similaritiesy in major emission sources for both sites. 271 The regional pollution was related to the special topography of the basin, which is a closed lowland 272 surrounded by high mountains on all sides (Fig. 1). The mean elevation in the basin is about 200-700 m, while the surrounded mountains are around a scope of 1000-3000 m elevation. The Tibetan Plateau lies 273 close to the western Sichuan Basin, with an elevation above 4000 m. Those uniqueSuch a Plateau-Basin 274 topography feature often forms a barrier for the dispersion of pollutants and causes air stagnation within 275 the basin, thereby <del>obviously</del> facilitating <del>a regional</del> scale pollution events in the Sichuan Basin. 276

277 72-h air masses back trajectories analysis (18:00 local time) showed that air masses reaching at CD 278 and CQ mainly originated from the local areas in the basin (Fig. S1), confirming the influence of the 279 high mountainous surroundingaround the basin and the possibility of forming regional pollution. 280 TheseOur results were consistent with those foundconducted in earlier studiesby in Chengdu and Chongqing (Tian et al., 2017; Liao et al., 2017) in Chengdu and Chongqing, which suggested that air 281 282 masses had short-range trajectoriesy and primarily originated from inside the Sichuan Basin, highlighting the impacts of the special topography on PM2.5 pollution. A similar case has also been 283 found elsewhere, such as The regional pollution of PM2.5 were also observed in Po Valley, Italy 284

[revised manuscript text omitted]

346 distribution of most major components in PM2.5 throughout the basin in all the seasons except summer.

- 347 3.2 PM2.5 formation mechanisms and geographical origins
- 348 **3.2.1** Pollution episodes and key chemical components
- 349 Pollution episodes during the campaign are highlighted with shaded areas in Fig. 5. Theose pollution
- 350 periods (PP) were defined as daily PM2.5 concentration being above the NAAQS guideline value of 75
- 351  $\mu g m^{-3}$ . Similarly, the days with PM2.5 concentration below 75  $\mu g m^{-3}$  were characterized to be clean
- 352 periods (CP). A total of seven pollution episodes were identified during the campaign at each site. There
- 353 were three long-lasting pollution episodes occurred simultaneously at the two sites on 23-27 October
- 354 2014, 7(8)-26 January, and 26-28 (29) April 2015., A total of 34 and 31 polluted days were counted at
- 355 CD and CQ site, respectively, accounting for 30.4% and 28.6% of the entire sampling days (112 days).
- 356 The number of polluted days at CD was 8, 21, 4 and 1 in autumn, winter, spring and summer,
- 357 accounting for 29.6%, 75%, 14.3% and 3.4% of the total sampling days in each season, respectively,
- 358 and at CQ they were 4, 19, 6 and 2, accounting for 14.8%, 67.9%, 21.4% and 6.9%.
- Stagnant atmosphere and high relative humidity were important factors causing PM2.5 pollution 359 events, as was found in this and earlier studies. (Zheng et al., 2015b; Chen et al., 2017; Liao et al., 2017). 360 Compared with the clean periods, the pollutioned periods were usually characterized by low planetary 361 362 boundary layer height and weak wind speed (Table S1), which suppress pollutants dispersion vertically and horizontally. Temperature increased during the long-lasting pollution episodes, which promoted 363 gas-to-particle transformation, forming secondary aerosols. RH remained high (68-88%) during 364 pollution episodes (except in spring at CQ), although not much different from clean periods, which was 365 also conducive for aqueous-phase reactions converting gaseous pollutants into aerosols (Chen et al., 366
- 367 2017;Tian et al., 2017).
- 368 To explore the major chemical pollutants responsible for polluted days, daily PM2.5 data were-
- 369 categorized into two groups, i.e. clean days and polluted days based on PM2.5 concentrations below and
- 370 above the NAAQS guideline value of 75  $\mu$ g m-3, respectively. 34 and 31 polluted days were counted at-
- 371 CD and CQ site, respectively, accounting for 30.4% and 28.6% of the entire sampling days. The number
- 372 of polluted days at CD was 8, 21, 4 and 1 in autumn, winter, spring and summer, accounting for 29.6%,
- 373 <del>75%, 14.3% and 3.4% of the total sampling days in each season, respectively, and at CQ they were 4, 19,</del>
- 374 6 and 2, accounting for 14.8%, 67.9%, 21.4% and 6.9%. Note that these numbers were only fromone-month (around 28 days) sampling in each season. Considering the similar meteorological conditions and pollution levels during autumn and spring, these two seasons were combined together and was referred to as the warm season, while winter was referred to as the cold season. Because of the
 very small number of polluted days in summer at both sites, the PM2.5 data in this season were not discussed.

380 Looking more closely at a regional-scale long-lasting pollution episode in winter at CD and CQ, 381 from 8 January to 26 January 2015, the concentrations of PM2.5 and major chemical componentssitions increased dramatically on during polluted days periods compared to with clean days periods (Fig. 56).-382 For example, PM2.5 concentrations were more than doubled three times higher on polluted days 383 periods during the entire sampling periods at both sites,. The with the two dominant groups of 384 385 components-in PM2.5, SNA and OC, werebeing 2.5-2.8 times higher on during polluted days periods inboth cold and warm seasons at CD and 1.7-2.7 times higher at CQ. . However, larger variations were 386 found at CQ with SNA and OC increased by 2.7 and 1.7 times, and OM by 3.4 and 2.1 times in the cold-387 388 and warm season, respectively. Thus, while tThe enhancement of SNA and OC on-during pollutioned 389 days periods were similar at CD, but OC increased much more than SNA at CQ, indicating some different contributing factors to the high PM2.5 pollution at the two sites. Pollutants accumulation under 390 stagnant meteorological conditions might be a main factor at CD based on the similar magnitudes of the 391 enhancements of PM2.5 and its dominant components, while additional processes should have increased 392 393 OC more than other components at CQ.

In the cold season, the The percentage contributions of SNA to PM2.5 were similar on-during clean
 and polluted daysperiods, 38-41% at both sitesCD and CQ (Fig. S1S3). However, the percentage
 contribution of OM to PM2.5 decreased from 30.1% on clean days to 27.5% on polluted days at CD, and

397 increased from 26.9% to 34.9% at CQ. A different pattern was seen in the warm season, with no-

398 significant variations of OM fractions in  $PM_{2.5}$  between the clean and polluted days at either site, but an

399 increased SNA contribution by around 7% at CD and decreased contribution by 14% at CQ.

400 Concentrations of the individual SNA species (SO4 $\frac{2}{2}$ , NO3 $\frac{1}{2}$  and NH4 $\frac{1}{2}$ ) increased by a factor of

401  $\frac{1.22.5-3.3 \text{ on polluted days compared towith clean days in all the cases (Fig. 56). But the percentage}{\frac{1.22.5}{1.2}}$

402 contributions differed among the species as  $NO_3^{\pm}$  increased and  $SO_4^{\pm}$  decreased on polluted days. The

403 percentage contributions of SNA and OM to  $PM_{2.5}$  discussed above were different from those found in

404 eastern coastal China and North China Plain, where considerable increases were found for SNA and

- decreases for OM on polluted days than clean days (Tan et al., 2009;Wang et al., 2015a;Quan et al.,
- 406 2014;Zhang et al., 2015;Zhang et al., 2016;Cheng et al., 2015). The polluted periods in eastern coastal
- 407 China and North China Plain were accompanied with sharp increases of RH, which would promote
- 408 aqueous-phase formation of secondary inorganic aerosols and resulted in rapid elevation of  $SO_4^{2-}$  and
- 409  $\underline{NO_3}^{=}$  concentrations (Zheng et al., 2015b;Zheng et al., 2015a;Zhao et al., 2013b;Li et al., 2017). In
- 410 contrast, RH remained high during clean or polluted periods in the present study, suggesting that high
- 411 RH might not be the driving force for the pollution episodes in Sichuan Basin. Another point that need
- 412 to be mentioned is that, local sources were the main contributors to the pollution episodes in Sichuan
- 413 basin while sources outside local regions frequently contributed to pollution episodes in eastern coastal
- 414 China and North China Plain through long/medium range transport (Gao et al., 2015;Hua et al.,
- 415 2015; Wang et al., 2015b). This emphasized again the unique characteristics of PM2.5 pollution in-
- 416 Sichuan Basin due to its particular topography and meteorological conditions.
- 417 Concentrations of the individual SNA species ( $SO_4^2$ ,  $NO_3$  and  $NH_4^+$ ) increased by a factor of
- 418 1.2-3.3 on polluted days compared to clean days in all the cases (Fig. 5). But the percentage-
- 419 contributions differed among the species as  $NO_3^{-}$  increased and  $SO_4^{-2^{-}}$  decreased on polluted days. The
- 420 concentration of FS increased slightly at CD (less than  $3 \mu \text{g m}^{-3}$ ) but significantly (from 5.4 on clean-
- 421 days to 14.7 μg m-3 on polluted days) at CQ in warm season. The percentage contribution of FS to-
- 422 PM2.5-reached 15.3% on polluted days at CQ in warm season.

**423 **3.2.2 Transformation mechanisms of secondary aerosols**

- 424 In most cases, Mmeteorological conditions, atmospheric chemical processes and long-range transport are all responsible for PM2.5 accumulation on polluted days (Zheng et al., 2015b). CO and EC are 425 426 directly emitted from combustion processes and are not very reactive, thus, their concentrations in the 427 air are strongly controlled by meteorological parameters within a relatively short period, which make them a good tracer that can be used for separating different dominant factors contributing to pollutants 428 accumulation (Zheng et al., 2015b;Zhang et al., 2015). In the present study, CO was chosen as a 429 reference pollutant species for investigating other pollutants of interest. CO concentration increased by 430 a factor of 1.8 at CD and 1.5 at CQ during polluted periods than clean periods. In comparison, other 431 chemical species (except NO2) increased by a factor of 1.8-3.3. A similar contrast between CO and 432 other pollutants was also seen in (Li et al., 2017). Similar diurnal variations were also found between 433
- 434 clean and polluted periods for CO (Fig. S4), suggesting no significant extra CO emission during

435 polluted periods. Thus, the increased concentrations of CO during polluted periods were primarily driven by meteorological conditions. Both CO and EC concentrations increased on polluted days (Fig. 5-436 and Fig. S2), suggesting the important role the meteorological condition played on PM2.5 accumulation. 437 As expected, very weak winds (less than 0.7 m s-1) were observed on polluted days, which hindered the 438 pollutants horizontal transport. CO can be considered as a reference pollutant species whose temporal-439 440 variations were mainly from the impact of meteorological conditions. Therefore, The the impact of 441 atmospheric physical processes other factors on other pollutants can thus be explored reduced by scaling the concentrations of other pollutants to that of CO, meanings that the impact of chemical reactions can 442 then be captured based on the ratio of other pollutants to CO concentrations. For example, PM2.5 was 443 enhanced by a factor of 2.0-2.7 on polluted days than clean days in the two seasons and at the twoboth 444 sites, but the CO-scaled PM2.5 (the ratio of PM2.5 to CO concentration) only showed an enhancement of 445 a factor of 1.56-1.8 (Fig. 67), and the latter values were likely from the enhanced secondary aerosol 446 formation. 447

448 As shown in Fig. 67, the CO-scaled SNA was 60-90% higher on polluted days with individual 449 species 40-120% higher (except in the warm season at CQ), even though their gaseous precursor (SO2) and NO2, no data for NH3) were only less than 30% higher. This suggests stronger chemical 450 transformation from gaseous precursors to particle formation on polluted days. Sulfur oxidation ratio 451  $(SOR = n-SO_4^{2-}/(n-SO_4^{2-}+n-SO_2))$  and nitrogen oxidation ratio  $(NOR = n-NO_3^{-}/(n-NO_3^{-}+n-NO_2))$  were 452 453 defined to evaluate the degree of secondary transformation (*n* refers to as the molar concentration) (Hu et al., 2014). NOR increased from 0.09 on clean days to 0.16 on polluted days at CD and from 0.07 to 454 0.14 at CQ. SOR increased only slightly, from 0.31 to 0.35 at CD and 0.28 to 0.35 at CQ. In the warm-455 456 season, NOR and SOR exhibited a similar pattern as those in the cold season with the exception of NOR 457 at CQ, which might be related to the high temperature in the warm season. The CO-scaled SOC 458 increased by a factor of 2.6 and 1.5 on polluted days in the cold and warm season at CQ, but no significant change or decrease was found at CD. Moreover, increased SOC/OC only occurred at CQ in-459 the cold season (Fig. 67). The different patterns in SOC (or SOC/OC) than SNA (or SOR and NOR) 460 461 suggests that secondary organic aerosols (SOA) production iwas of less important than SNA production 462 in most occasions except in the cold season at CQCD.

463  $\underline{SO_4}^{2-}$  is predominantly formed via homogeneous gas-phase oxidation. In this pathway,  $\underline{SO_2}$  is 464 firstly oxidized by OH radical to SO\_3, and then to H2SO4 (Stockwell and Calvert, 1983;Blitz et al.,

| 465                                                                                                   | 2003). Apart from homogeneous reaction, particulate $SO_4^{\frac{2}{2}}$ can also be formed through heterogeneous                                                                                                                                                                                                                                                                                                                                                                                                                                                                                                                                                                                                                                                                                                                                                                                                                                                                                                                                                                                                                                                                                                                                                                                                                                                                                                                                                                                                             |
|-------------------------------------------------------------------------------------------------------|-------------------------------------------------------------------------------------------------------------------------------------------------------------------------------------------------------------------------------------------------------------------------------------------------------------------------------------------------------------------------------------------------------------------------------------------------------------------------------------------------------------------------------------------------------------------------------------------------------------------------------------------------------------------------------------------------------------------------------------------------------------------------------------------------------------------------------------------------------------------------------------------------------------------------------------------------------------------------------------------------------------------------------------------------------------------------------------------------------------------------------------------------------------------------------------------------------------------------------------------------------------------------------------------------------------------------------------------------------------------------------------------------------------------------------------------------------------------------------------------------------------------------------|
| 466                                                                                                   | reactions with dissolved $O_3$ or $H_2O_2$ under the catalysis of transition metal and in-cloud process                                                                                                                                                                                                                                                                                                                                                                                                                                                                                                                                                                                                                                                                                                                                                                                                                                                                                                                                                                                                                                                                                                                                                                                                                                                                                                                                                                                                                       |
| 467                                                                                                   | (Ianniello et al., 2011). HNO 3 is primarily produced from the reactions between NO 2 and OH radical                                                                                                                                                                                                                                                                                                                                                                                                                                                                                                                                                                                                                                                                                                                                                                                                                                                                                                                                                                                                                                                                                                                                                                                                                                                                                                                                                                                                    |
| 468                                                                                                   | during the daytime and later combines with NH 3 to produce particulate NO 3 2 (Calvert and Stockwell,                                                                                                                                                                                                                                                                                                                                                                                                                                                                                                                                                                                                                                                                                                                                                                                                                                                                                                                                                                                                                                                                                                                                                                                                                                                                                                                                                                                        |
| 469                                                                                                   | 1983). Particulate <math>NO_3^2</math> can also be formed through heterogeneous hydrolysis of <math>N_2O_5</math> on moist and                                                                                                                                                                                                                                                                                                                                                                                                                                                                                                                                                                                                                                                                                                                                                                                                                                                                                                                                                                                                                                                                                                                                                                                                                                                                                                                                                                                         |
| 470                                                                                                   | acidic aerosols during nighttime (Ravishankara, 1997;Brown and Stutz, 2012). SO4 2- and NO3 - -                                                                                                                                                                                                                                                                                                                                                                                                                                                                                                                                                                                                                                                                                                                                                                                                                                                                                                                                                                                                                                                                                                                                                                                                                                                                                                                                                                                                         |
| 471                                                                                                   | (Stockwell and Calvert, 1983; Blitz et al., 2003; Calvert and Stockwell, 1983), heterogeneous reactions-                                                                                                                                                                                                                                                                                                                                                                                                                                                                                                                                                                                                                                                                                                                                                                                                                                                                                                                                                                                                                                                                                                                                                                                                                                                                                                                                                                                                                      |
| 472                                                                                                   | might also be important formation mechanisms for these SNA species (Quan et al., 2015; Zheng et al.,                                                                                                                                                                                                                                                                                                                                                                                                                                                                                                                                                                                                                                                                                                                                                                                                                                                                                                                                                                                                                                                                                                                                                                                                                                                                                                                                                                                                                          |
| 473                                                                                                   | 2015a; Zhao et al., 2013b). Similarly, SOA is mainly formed through photochemical oxidation of                                                                                                                                                                                                                                                                                                                                                                                                                                                                                                                                                                                                                                                                                                                                                                                                                                                                                                                                                                                                                                                                                                                                                                                                                                                                                                                                                                                                                                |
| 474                                                                                                   | primary VOCs followed by condensation of SVOC onto particles as well as through aqueous-phase                                                                                                                                                                                                                                                                                                                                                                                                                                                                                                                                                                                                                                                                                                                                                                                                                                                                                                                                                                                                                                                                                                                                                                                                                                                                                                                                                                                                                                 |
| 475                                                                                                   | reactions (Ervens et al., 2011). While photochemical reactions are mostly influenced by temperature                                                                                                                                                                                                                                                                                                                                                                                                                                                                                                                                                                                                                                                                                                                                                                                                                                                                                                                                                                                                                                                                                                                                                                                                                                                                                                                                                                                                                           |
| 476                                                                                                   | and oxidants amount, heterogeneous reactions always depends on RH. To further explore the formation                                                                                                                                                                                                                                                                                                                                                                                                                                                                                                                                                                                                                                                                                                                                                                                                                                                                                                                                                                                                                                                                                                                                                                                                                                                                                                                                                                                                                           |
| 477                                                                                                   | mechanisms of secondary aerosols, -SOR, NOR and SOC/OC data were grouped with temperature (at                                                                                                                                                                                                                                                                                                                                                                                                                                                                                                                                                                                                                                                                                                                                                                                                                                                                                                                                                                                                                                                                                                                                                                                                                                                                                                                                                                                                                                 |
| 478                                                                                                   | 2°C interval), RH (at 5% interval) and daytime O 3 concentration (at 5 μg m -3 interval) bins (Fig. 8). An                                                                                                                                                                                                                                                                                                                                                                                                                                                                                                                                                                                                                                                                                                                                                                                                                                                                                                                                                                                                                                                                                                                                                                                                                                                                                                                                                                                              |
| 479                                                                                                   | obvious increase of SOR with increasing RH was found at both sites, but this was no the case for                                                                                                                                                                                                                                                                                                                                                                                                                                                                                                                                                                                                                                                                                                                                                                                                                                                                                                                                                                                                                                                                                                                                                                                                                                                                                                                                                                                                                              |
|                                                                                                       |                                                                                                                                                                                                                                                                                                                                                                                                                                                                                                                                                                                                                                                                                                                                                                                                                                                                                                                                                                                                                                                                                                                                                                                                                                                                                                                                                                                                                                                                                                                               |
| 480                                                                                                   | temperature and O 3 concentration, suggesting heterogeneous processes played an-important roles in the                                                                                                                                                                                                                                                                                                                                                                                                                                                                                                                                                                                                                                                                                                                                                                                                                                                                                                                                                                                                                                                                                                                                                                                                                                                                                                                                                                                                             |
| 480
481                                                                                            | temperature and O3 concentration, suggesting heterogeneous processes played an important roles in the
formation of SO42-. Interestingly, SOR exhibited a decreasinged trend with increasing O3 concentration                                                                                                                                                                                                                                                                                                                                                                                                                                                                                                                                                                                                                                                                                                                                                                                                                                                                                                                                                                                                                                                                                                                                                                                                                                                     |
| 480
481
482                                                                                     | temperature and $O_3$ concentration, suggesting heterogeneous processes played an-important roles in the formation of $SO_4^{2^-}$ . Interestingly, SOR exhibited a decreasinged trend with increasing $O_3$ concentration at $O_3$ concentrations lower than 15 µg m -3 and an opposite trend was found at $O_3$ concentrations above                                                                                                                                                                                                                                                                                                                                                                                                                                                                                                                                                                                                                                                                                                                                                                                                                                                                                                                                                                                                                                                                                                                                                                             |
| 480
481
482
483                                                                              | temperature and $O_3$ concentration, suggesting heterogeneous processes played an-important roles in the formation of $SO_4^{2^-}$ . Interestingly, SOR exhibited a decreasinged trend with increasing $O_3$ concentration at $O_3$ concentrations lower than 15 µg m -3 and an opposite trend was found at $O_3$ concentrations above $20 \mu g m^{-3}$ (Fig. 8). Additionally, high PM 2.5 concentrations were mostly associated with lower $O_3$ .                                                                                                                                                                                                                                                                                                                                                                                                                                                                                                                                                                                                                                                                                                                                                                                                                                                                                                                                                                                                                                                   |
| 480
481
482
483
484                                                                       | temperature and O 3 concentration, suggesting heterogeneous processes played <del>an</del> -important roles in the
formation of SO 4 2- . Interestingly, SOR exhibited a decreasing <del>ed</del> trend with increasing O 3 concentration
at O 3 concentrations lower than 15 μg m -3 and an opposite trend was found at O 3 concentrations above
20 μg m -3 (Fig. 8). Additionally, high PM 2.5 concentrations were mostly associated with lower O 3
concentrations. Aerosols are generally considered as a constraining factor to O 3 production due to their.                                                                                                                                                                                                                                                                                                                                                                                                                                                                                                                                                                                                                                                                                                                                                                                                                         |
| 480
481
482
483
484
485                                                                | temperature and O 3 concentration, suggesting heterogeneous processes played <del>an</del> -important roles in the
formation of SO 4 2- . Interestingly, SOR exhibited a decreasing <del>ed</del> trend with increasing O 3 concentration
at O 3 concentrations lower than 15 μg m -3 and an opposite trend was found at O 3 concentrations above
20 μg m -3 (Fig. 8). Additionally, high PM 2.5 concentrations were mostly associated with lower O 3
concentrations. Aerosols are generally considered as a constraining factor to O 3 production due to their
absorption and scattering of UV radiation, which reduce solar radiation and consequently decrease                                                                                                                                                                                                                                                                                                                                                                                                                                                                                                                                                                                                                                                                                                                     |
| 480

486                                                         | temperature and O 3 concentration, suggesting heterogeneous processes played <del>an</del> -important roles in the
formation of SO 4 2- . Interestingly, SOR exhibited a decreasing <del>ed</del> trend with increasing O 3 concentration
at O 3 concentrations lower than 15 μg m -3 and an opposite trend was found at O 3 concentrations above
20 μg m -3 (Fig. 8). Additionally, high PM 2.5 concentrations were mostly associated with lower O 3
concentrations. Aerosols are generally considered as a constraining factor to O 3 production due to their
absorption and scattering of UV radiation, which reduce solar radiation and consequently decrease
photochemical activity. Apparently, other factors could become dominant in O 3 production and alter                                                                                                                                                                                                                                                                                                                                                                                                                                                                                                                                                                                                   |
| 480

486                                                  | temperature and $O_3$ concentration, suggesting heterogeneous processes played an-important roles in the
formation of $SO_4^{2^2}$ . Interestingly, SOR exhibited a decreasinged trend with increasing $O_3$ concentration
at $O_3$ concentrations lower than 15 µg m -3 and an opposite trend was found at $O_3$ concentrations above
$20 µg m^{-3}$ (Fig. 8). Additionally, high PM 2.5 concentrations were mostly associated with lower $O_3$
concentrations. Aerosols are generally considered as a constraining factor to $O_3$ production due to their
absorption and scattering of UV radiation, which reduce solar radiation and consequently decrease
photochemical activity. Apparently, other factors could become dominant in $O_3$ production and alter
such a trend. Low $O_3$ concentrations were further found to be associated with high RH (Fig. S5),                                                                                                                                                                                                                                                                                                                                                                                                                                                                                                                                                                                                            |
| 480

488                                    | temperature and $O_3$ concentration, suggesting heterogeneous processes played an-important roles in theformation of $SO_4^{2^{2}}$ . Interestingly, SOR exhibited a decreasinged trend with increasing $O_3$ concentrationat $O_3$ concentrations lower than 15 µg m -3 and an opposite trend was found at $O_3$ concentrations above20 µg m -3 (Fig. 8). Additionally, high PM 2.5 concentrations were mostly associated with lower $O_3$ .concentrations. Aerosols are generally considered as a constraining factor to $O_3$ production due to theirabsorption and scattering of UV radiation, which reduce solar radiation and consequently decreasephotochemical activity. Apparently, other factors could become dominant in $O_3$ production and altersuch a trend. Low $O_3$ concentrations were further found to be associated with high RH (Fig. S5),indicating that the formation of $SO_4^{2^{2}}$ during the polluted periods was dominated by heterogeneous                                                                                                                                                                                                                                                                                                                                                                                                                                                                                                                   |
| 480

489                             | temperature and $O_3$ concentration, suggesting heterogeneous processes played an-important roles in theformation of $SO_4^{2^2}$ . Interestingly, SOR exhibited a decreasinged trend with increasing $O_3$ concentrationat $O_3$ concentrations lower than 15 µg m -3 and an opposite trend was found at $O_3$ concentrations above $20 µg m^{-3}$ (Fig. 8). Additionally, high PM 2.5 concentrations were mostly associated with lower $O_3$ .concentrations. Aerosols are generally considered as a constraining factor to $O_3$ production due to theirabsorption and scattering of UV radiation, which reduce solar radiation and consequently decreasephotochemical activity. Apparently, other factors could become dominant in $O_3$ production and altersuch a trend. Low $O_3$ concentrations were further found to be associated with high RH (Fig. S5),indicating that the formation of $SO_4^{2^2}$ during the polluted periods was dominated by heterogeneousaqueous processes rather than photochemical reactions at both sites, as suggested in many previous.                                                                                                                                                                                                                                                                                                                                                                                                                          |
| 480

490                             | temperature and $O_3$ concentration, suggesting heterogeneous processes played an-important roles in the
formation of $SO_4^{2^2}$ . Interestingly, SOR exhibited a decreasinged trend with increasing $O_3$ concentration
at $O_3$ concentrations lower than 15 µg m -3 and an opposite trend was found at $O_3$ concentrations above
$20 µg m^{-3}$ (Fig. 8). Additionally, high PM 2.5 concentrations were mostly associated with lower $O_3$ .
concentrations. Aerosols are generally considered as a constraining factor to $O_3$ production due to their
absorption and scattering of UV radiation, which reduce solar radiation and consequently decrease
photochemical activity. Apparently, other factors could become dominant in $O_3$ production and alter
such a trend. Low $O_3$ concentrations were further found to be associated with high RH (Fig. S5),
indicating that the formation of $SO_4^{2^2}$ during the polluted periods was dominated by heterogeneous
aqueous processes rather than photochemical reactions at both sites, as suggested in many previous.
studies (Quan et al., 2015;Zheng et al., 2015a;Zhao et al., 2013b)                                                                                                                                                                                                                                                                                                                 |
| 480

491               | temperature and $O_3$ concentration, suggesting heterogeneous processes played an-important roles in the
formation of $SO_4^{2^2}$ . Interestingly, SOR exhibited a decreasinged trend with increasing $O_3$ concentration
at $O_3$ concentrations lower than 15 µg m -3 and an opposite trend was found at $O_3$ concentrations above
$20 µg m^{-3}$ (Fig. 8). Additionally, high PM 2.5 concentrations were mostly associated with lower $O_3$
concentrations. Aerosols are generally considered as a constraining factor to $O_3$ production due to their
absorption and scattering of UV radiation, which reduce solar radiation and consequently decrease
photochemical activity. Apparently, other factors could become dominant in $O_3$ production and alter
such a trend. Low $O_3$ concentrations were further found to be associated with high RH (Fig. S5),
indicating that the formation of $SO_4^{2^2}$ during the polluted periods was dominated by heterogeneous
aqueous processes rather than photochemical reactions at both sites, as suggested in many previous
studies (Quan et al., 2015;Zheng et al., 2015a;Zhao et al., 2013b)                                                                                                                                                                                                                                                                                                                    |
| 480

492        | temperature and $O_3$ concentration, suggesting heterogeneous processes played an-important roles in the
formation of $SO_4^{2^2}$ . Interestingly, SOR exhibited a decreasinged trend with increasing $O_3$ concentration.
at $O_3$ concentrations lower than 15 µg m -3 and an opposite trend was found at $O_3$ concentrations above
$20 µg m^{-3}$ (Fig. 8). Additionally, high $PM_{2.5}$ concentrations were mostly associated with lower $O_3$
concentrations. Acrosols are generally considered as a constraining factor to $O_3$ production due to their
absorption and scattering of UV radiation, which reduce solar radiation and consequently decrease
photochemical activity. Apparently, other factors could become dominant in $O_3$ production and alter
such a trend. Low $O_3$ concentrations were further found to be associated with high RH (Fig. S5),
indicating that the formation of $SO_4^{2^2}$ during the polluted periods was dominated by heterogeneous
aqueous processes rather than photochemical reactions at both sites, as suggested in many previous.
studies (Quan et al., 2015;Zheng et al., 2015a;Zhao et al., 2013b)
Unlike SOR, NOR increased with both temperature and RH, suggesting combined the effects from
homogeneous and heterogeneous reactions. However, under the very high temperature and RH.                                                                                                                            |
| 480

493 | temperature and $O_3$ concentration, suggesting heterogeneous processes played an-important roles in the.
formation of $SO_4^{2^2}$ . Interestingly, SOR exhibited a decreasinged trend with increasing $O_3$ concentration.
at $O_3$ concentrations lower than 15 µg m 23 and an opposite trend was found at $O_3$ concentrations above.
20 µg m 23 (Fig. 8). Additionally, high PM 2.5 concentrations were mostly associated with lower $O_3$ .
concentrations. Aerosols are generally considered as a constraining factor to $O_3$ production due to their.
absorption and scattering of UV radiation, which reduce solar radiation and consequently decrease
photochemical activity. Apparently, other factors could become dominant in $O_3$ production and alter.
such a trend. Low $O_3$ concentrations were further found to be associated with high RH (Fig. S5),
indicating that the formation of $SO_4^{2^2}$ during the polluted periods was dominated by heterogeneous.
aqueous processes rather than photochemical reactions at both sites, as suggested in many previous.
studies (Quan et al., 2015;Zheng et al., 2015a;Zhao et al., 2013b)
Unlike SOR, NOR increased with both temperature and RH, suggesting combined the effects from
homogeneous and heterogeneous reactions. However, under the very high temperature and RH.
conditions, NOR exhibited a decreasing trend with increasing temperature and RH. Volatilization of |

495 about the cause of the decreasing trend of NOR under high RH.

[revised manuscript text omitted]

- 550 formation mechanism of secondary aerosols on moderate polluted days (high O3 levels and SR, low RH)
- 551 while heterogeneous reactions likely played a more important role on heavy polluted days (low O3-
- 552 levels and slightly higher RH). This hypothesis needs further verification using high-resolution data.

**553 **3.4.3 Impact of NH3 amount and RH on NO3- concentrations-**

In Sect. 3.4.2, it was found that the CO-scaled NO3- increased dramatically in the cold season butdecreased in the warm season on polluted days at CQ. To explain the different season patterns, majorfactors affecting NO3- are explored, including NH3 levels and RH. The neutralization ratio (NR) isdefined as -

558
$$\frac{NR = \frac{[NH_4^+]}{2[SO_4^{2-}] + [NO_3^-]}}{(4)}$$

[revised manuscript text omitted]

---

## Author Response (AR2)

Dear Editor:

We have carefully revised the manuscript and addressed all of the comments provided by the reviewer. The details can be found in our enclosed responses to the reviewer's comments. For your and the reviewer's convenience in reviewing the changes, a copy of the paper with track-changes is also attached below.

Thank you for taking care of the review process for this paper.

Sincerely,

Huanbo Wang and co-authors

We greatly appreciate the helpful comments from the reviewer, which have helped us improve the paper. We have addressed all of the comments carefully, as detailed below. Our responses start with "R:".

Sichuan Basin is one of the regions in China with severe aerosol pollution, while the chemical component analysis is still rare. This study fills the gap in this area. Under this condition, the short-term (four selected months) and low time resolution sampling is still acceptable. Thus, I suggest it published after major revision.
In general, the manuscript is not well organized. Authors should put related results together to support one conclusion. The results now are presented with several short paragraphs which are not carefully ordered, e.g., section 3.1.2. From line 197 to line 234, it shows the results of SNA, SNA, carbonaceous aerosols, and SNA again. The readers can hardly find the sufficient information to know and understand the results. Presentation should be modified by native speaker before publication.

**R:** We have better organized sections 3.1.1, 3.1.2, 3.1.3, and 3.2.1 following this comment. Several short paragraphs in sections 3.1.1 and 3.1.2 were combined based on their contents. Six short paragraphs in section 3.1.3 were reorganized into four paragraphs following the order of similarity, difference (SNA and carbonaceous component) and correlation analysis between CD and CQ sampling site. Section 3.2.1 was also integrated according to the pollution episodes during the one-year sampling periods and key chemical compositions caused $PM_{2.5}$ pollution in winter.

We have rewritten section 3.1.2 following this order: we first summarized the seasonal variations of $PM_{2.5}$ concentrations and the impacts of the meteorological parameters on the seasonal trends. We then explored the seasonal trends of major components of $PM_{2.5}$ with discussions for the same chemical components grouped into one place, as recommended by the reviewer.

In addition, our discussion basically kept the same sequence for the major chemical components groups: we first discussed the fine soil due to their different seasonal trends and spatial variations at CD and CQ, then SNA and individual $SO_4^{2-}$, $NO_3^-$, $NH_4^+$; and finally, carbonaceous components including organic carbon and elemental carbon.

We have also carefully proofread the revised paper to minimize any grammar errors, and have asked a native speaker to have a final check of the paper.

There are many qualitative speculations in the article. Without robust evidence, to do such speculations is unreliable and may even mislead readers. Many conclusions are not based on strict logic.
Line 203: "The majority of $PM_{2.5}$ components showed a summer minimum, which was caused by high planetary boundary layer height favoring pollutants dispersion and abundant precipitation favoring wet scavenging." I think it will be more appropriate to show precipitation to support this statement.

**R:** We have added quantitative discussion wherever possible. For example, we have added seasonal precipitation amount in Table 1 and related discussions. The precipitation amount was the highest in summer at both CD and CQ, which supported our conclusion. We admit that there are still places with qualitative discussions due to the nature of the study, the lack of data or the limitation for the current knowledge on this topic.

Line 208 and 217. The statement could not only be supported by other references. Other mechanisms can also lead to similar or opposite results. More observed results are needed.

**R:** We have added the observed results such as gaseous precursors to explain the seasonal trends of major chemical components in $PM_{2.5}$. Detailed discussions are included in section 3.1.2 in the revised manuscript.

Line 237: "Although none of the two sites alone can represent the whole region of the Sichuan basin, the similarities in the characteristics of the major pollutants between the two sites should represent the regional-scale characteristics of urban-environment pollution while the differences between the two sites should reflect the sub-regional characteristics of urban pollution." As the authors response to previous suggestions, "The two monitoring sites selected in this study should represent the typical urban environment in their respective cities". Thus, many similarities reflect the characteristics of urban region instead of the whole region.

**R:** Yes, the selected sampling sites at CD and CQ represent the typical urban environment. We have changed the sentence "a moderate similarity was observed in autumn, winter and spring on the basis of low COD values (0.15-0.18), indicating the regional-scale $PM_{2.5}$ pollution pattern in Sichuan Basin" to "a moderate similarity was observed in autumn, winter and spring on the basis of low COD values (0.15-0.18), indicating limited differences between the two urban environments in the Sichuan Basin".

Line 403, the interaction between aerosol and ozone are quite complicated. Aerosols can provide an interface for the heterogeneous reaction of ozone products. Previous study shows that this mechanism usually contributes more than the impact through photochemical process. Fig S5 also could not support that "the formation of $SO_4^{2-}$ during the polluted periods was dominated by heterogeneous aqueous processes rather than photochemical reactions".

**R:** Yes, we agree with the reviewer's comments and added the explanation on why high $PM_{2.5}$ concentrations associated with lower $O_3$ concentrations: On the one hand, aerosols are generally considered as a constraining factor to $O_3$ production due to their absorption and scattering of UV radiation, which reduce solar radiation and consequently decrease photochemical activity. On the other hand, aerosols can provide an interface for the heterogeneous reaction, in accordance with $O_3$ consumption and secondary aerosol formation, which would result in decreasing $O_3$ concentrations and increasing $PM_{2.5}$ concentrations.

Fig. S5 could not give a solid evidence for the formation mechanism of $SO_4^{2-}$. However, it can be found that the ambient RH remained high at low $O_3$ concentrations, which was beneficial to form $SO_4^{2-}$ through heterogeneous aqueous processes, resulting high SOR value at high RH accompanied with low $O_3$ concentrations.

Line 365, "Similar diurnal variations were also found between clean and polluted periods for CO (Fig. S4), suggesting no significant extra CO emission during polluted periods." Is that possible to give a quantitative value to support that the variations are similar? Also, I don't consider this can support no extra CO emission directly.

**R:** The diurnal variations of CO between clean and pollution periods were significant correlation with Pearson coefficient of 0.83 and 0.50 at both CD and CQ ($p<0.05$) based on statistical analysis. As suggested by the reviewer, it also could not support the statement that no extra CO emission occurred only according to the diurnal variations of CO. Thus, we deleted those explanations in the revised manuscript. Nevertheless, in practice scaling the concentrations of other pollutants to that of CO (or BC) was also an effective method to reduce the impact of atmospheric physical processes on other pollutants, which has been used widely in the previous study (Zhang et al., 2015; Zheng et al., 2015; Liggio et al., 2016; Hu et al., 2013), and we added those citations in the revised manuscript.

Zhang, Q., Quan, J. N., Tie, X. X., Li, X., Liu, Q., Gao, Y., and Zhao, D. L.: Effects of meteorology and secondary particle formation on visibility during heavy haze events in Beijing, China, Sci Total Environ, 502, 578-584, 10.1016/j.scitotenv.2014.09.079, 2015.
Zheng, G. J., Duan, F. K., Su, H., Ma, Y. L., Cheng, Y., Zheng, B., Zhang, Q., Huang, T., Kimoto, T., Chang, D., Poschl, U., Cheng, Y. F., and He, K. B.: Exploring the severe winter haze in Beijing: the impact of synoptic weather, regional transport and heterogeneous reactions, Atmos Chem Phys, 15, 2969-2983, 10.5194/acp-15-2969-2015, 2015b.
Liggio, J., Li, S.M., Hayden, K., Taha, Y.M., Stroud, C., Darlington, A., Drollette, B.D., Gordon, M., Lee, P., Liu, P., Leithead, A., Moussa, S.G., Wang, D., O'Brien, J., Mittermeier, R.L., Brook, J.R., Lu, G., Staebler, R.M., Han, Y.M., Tokarek, T.W., Osthoff, H.D., Makar, P.A., Zhang, J.H., Plata, D.L., Gentner, D.R., 2016. Oil sands operations as a large source of secondary organic aerosols. Nature 534, 91-94.
Hu, W.W., Hu, M., Yuan, B., Jimenez, J.L., Tang, Q., Peng, J.F., Hu, W., Shao, M., Wang, M., Zeng, L.M., Wu, Y.S., Gong, Z.H., Huang, X.F., He, L.Y., 2013. Insights on organic aerosol aging and the influence of coal combustion at a regional receptor site of central eastern China. Atmospheric Chemistry and Physics 13, 10095-10112.

Some speculations are even self-contradictory. Line 208: "It is also noted that the seasonal variations of $NO_3$ were much larger than those of $SO_4$ and $NH_4$. This can be explained by the enhanced formation of $NO_3$ under high RH in winter, and volatility of $NH_4NO_3$ in summer under high temperature condition." And Line 348: "In contrast, RH remained high during clean or polluted periods in the present study, suggesting that high RH might not be the driving force for the pollution episodes in Sichuan Basin." Also, it is not convinced to claim RH might not be the driving force supported by this. Many factors including meteorological conditions and emission are combined together to form polluted episodes.

**R:** Line 208 and line 348 is not self-contradictory. As shown in Table 1, the average ambient RH in winter was 12% higher than that in summer. Therefore, compared with that in summer, the increased ambient RH in winter would enhance the formation of $NO_3^-$ through the heterogeneous aqueous processes.

However, the average ambient RH during clean period and pollution period in winter was 81.9% and 80.2% at CD, and 70.2% and 68.0% at CQ, respectively (Table S1). No significant variations of ambient RH between clean period and pollution period were observed (less than 3%). Thus, the high ambient RH during the whole sampling period in winter was beneficial to the $NO_3^-$ formation, but it might not be the primary cause of the dramatic increase of $PM_{2.5}$ concentrations during the pollution period.

The conclusions part are all qualitative even without a number.

**R:** We feel that Conclusion should be rewritten differently from the Abstract. In the Abstract quantitative statement summarizing major results should be presented while in the Conclusion section, results not covered in the abstract, uncertainty discussion, future research recommendations should be presented. However, we have added some quantitative statement following this comment.

Besides, I have some specific comments on the manuscript as follows:
1. Line 238, "regional" instead of "reginal".

**R:** Corrected.

2. Line 453, It is better for understanding if "Neijiang, Zigong, Yibin and Luzhou" could be marked in Fig 1.

**R:** We have added the mentioned cities in Fig 1.

3. The period should be stated in caption of Fig 6 and Fig 7.

**R:** We have added the clean period (CP) and pollution period (PP) in the caption of Fig 6 and Fig 7. 
[revised manuscript text omitted]